# Sodium Thiosulfate: An Innovative Multi-Target Repurposed Treatment Strategy for Late-Onset Alzheimer’s Disease

**DOI:** 10.3390/ph17121741

**Published:** 2024-12-23

**Authors:** Melvin R. Hayden, Neetu Tyagi

**Affiliations:** 1Department of Internal Medicine, Endocrinology Diabetes and Metabolism, Diabetes and Cardiovascular Disease Center, University of Missouri School of Medicine, One Hospital Drive, Columbia, MO 65211, USA; 2Department of Physiology, University of Louisville School of Medicine, Louisville, KY 40202, USA; neetu.tyagi@louisville.edu

**Keywords:** Alzheimer’s disease, chelation, hydrogen sulfide, nitric oxide, oxidative stress, reactive oxygen species, sodium thiosulfate

## Abstract

Late-onset Alzheimer’s disease (LOAD) is a chronic, multifactorial, and progressive neurodegenerative disease that associates with aging and is highly prevalent in our older population (≥65 years of age). This hypothesis generating this narrative review will examine the important role for the use of sodium thiosulfate (STS) as a possible multi-targeting treatment option for LOAD. Sulfur is widely available in our environment and is responsible for forming organosulfur compounds that are known to be associated with a wide range of biological activities in the brain. STS is known to have (i) antioxidant and (ii) anti-inflammatory properties; (iii) chelation properties for calcium and the pro-oxidative cation metals such as iron and copper; (iv) donor properties for hydrogen sulfide production; (v) possible restorative properties for brain endothelial-cell-derived bioavailable nitric oxide. Thus, it becomes apparent that STS has the potential for neuroprotection and neuromodulation and may allow for an attenuation of the progressive nature of neurodegeneration and impaired cognition in LOAD. STS has been successfully used to prevent cisplatin oxidative-stress-induced ototoxicity in the treatment of head and neck and solid cancers, cyanide and arsenic poisoning, and fungal skin diseases. Most recently, intravenous STS has become part of the treatment plan for calciphylaxis globally due to vascular calcification and ischemia-induced skin necrosis and ulceration. Side effects have been minimal with reports of metabolic acidosis and increased anion gap; as with any drug treatment, there is also the possibility of allergic reactions, possible long-term osteoporosis from animal studies to date, and minor side-effects of nausea, headache, and rhinorrhea if infused too rapidly. While STS poorly penetrates the intact blood–brain barrier(s) (BBBs), it could readily penetrate BBBs that are dysfunctional and disrupted to deliver its neuroprotective and neuromodulating effects in addition to its ability to penetrate the blood–cerebrospinal fluid barrier of the choroid plexus. Novel strategies such as the future use of nano-technology may be helpful in allowing an increased entry of STS into the brain.

## 1. Introduction

Sporadic, non-genetic, late-onset Alzheimer’s disease (LOAD) is a chronic, age-related (≥65 years of age), multifactorial, progressive, conformational, and neurodegenerative disease as compared to the genetic-related early-onset Alzheimer’s disease (EOAD) [1,2,3]. Importantly, those individuals ≥65 years of age have a prevalence of 19–30%, and approximately 10.5% also have a lifetime risk of developing LOAD [4]. Approximately 5 million Americans aged 65 and older were diagnosed with LOAD and related dementias in 2014, and that number is expected to more than double to 13.9 million by 2060 in the United States [5]. The post-world-war II baby boom generation has played an important role since they are still turning 65 at a rate of 10,000/day until 2030. This similar aging phenomenon is occurring globally and will contribute to our oldest-living population in history, which will continue to contribute to increasing aging-related diseases, including LOAD [2]. Indeed, LOAD is the leading worldwide cause of dementia [6].

LOAD may be characterized by the pathological accumulation of extracellular matrix amyloid beta (Aβ), neuritic plaques, and intracellular hyperphosphorylated misfolded proteins resulting in neurofibrillary tangles (NFTs) and associates with multiple hypotheses (Box 1) [2,7,8,9,10].

Box 1Major existing hypotheses and emerging concepts for late-onset Alzheimer’s disease (LOAD). Note that III. (Aβ cascade hypothesis) instigates IV. (tau hypothesis) (downward open arrow) and that there exists a vicious cycle between V. (inflammation) and VI. (oxidative redox stress via a downward and upward arrows) hypothesis that merges with VII. (the mitochondrial cascade hypothesis). Also, note that hypotheses V., VI., and VII. instigate the misfolded proteins III. and IV. Interestingly, these multiple hypotheses often interact with and influence one another. ***cns***CC, central nervous system cytokines and chemokines; ***p***CC, peripheral cytokines and chemokines; VCID, vascular contributions to cognitive impairment and dementia.

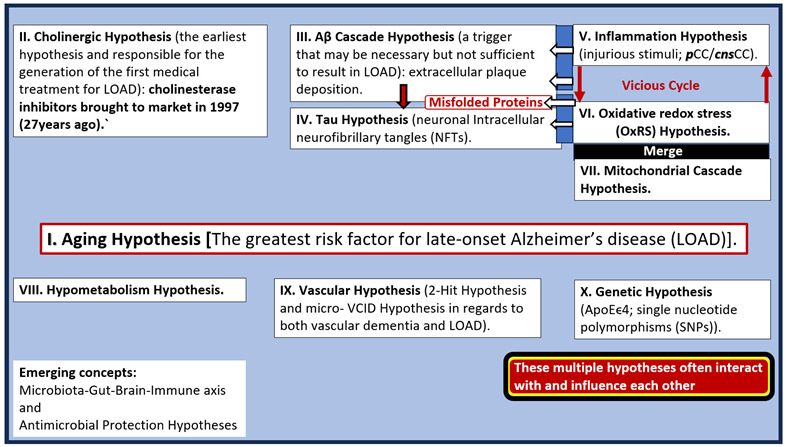



There still exists controversy regarding the Aβ amyloid cascade hypothesis (number III. in Box 1) proposed by Selkoe and Hardy [11,12]. However, multiple global laboratories and clinics still support the concept that there is an imbalance between the production and clearance of Aβ peptides and that Aβ peptides (especially the oligomeric forms) are an early and often the initiating factor for the development of LOAD [13]. Also, Musiek and Holtzman have commented that, even though Aβ may be necessary, it may not be sufficient to induce LOAD [8]. Still, the amyloid cascade hypothesis remains the dominant model of LOAD pathogenesis and guides the development of potential treatments [13]. Indeed, these multiple hypotheses as presented in Box 1 certainly support the concept that LOAD is a multifactorial, heterogenous disease that presents a dilemma in treatment when that treatment is for just one of these hypotheses. Multiple food and drug administration (FDA)-approved treatments over the years include (1) cholinesterase inhibitors such as donepezil and galantamine; (2) glutamate inhibitors such as memantine; (3) anti-amyloid therapy—monoclonal antibodies (mabs) such as lecanemab, aducanumab, donanemab-agbt, and ponezumab have been approved but are still being clinically studied in various trials; (4) various antipsychotics to modify impaired behavior and psychological symptoms such as brexpiprazole and newer insomnia medications including suvorexant [14,15].

Currently, there is an ongoing interest in repurposing existing medications that are approved by the United States Food and Drug Administration (FDA) and medications that treat not just one but several mechanisms (multi-target medicines) for the development of synaptic dysfunction, neurodegeneration, impaired cognition, and LOAD (Figure 1) [16,17,18].

Recently, the author has proposed that there exists a dementia quartet consisting of (1). oxidative-redox stress (OxRS); (2). neuroinflammatory; (3). neurovascular; (4). neurodegenerative mechanisms that associate with small vessel disease (SVD) that are present in LOAD (Figure 2) [10].

The multifactorial mechanisms of the dementia quartet, multiple hypotheses in Box 1, and multiple targets in Figure 1 are not likely to respond to a treatment that only addresses just one of these mechanisms or hypotheses [18]. Therefore, a multi-targeted therapeutic program that simultaneously targets multiple factors may be more effective than a single-targeting drug approach that has been previously utilized in the past [19]. Indeed, a personalized, multi-therapeutic program based on an individual’s genetics and biochemistry may be preferable over a single-drug/mono-therapeutic approach. Currently, there are no mono-therapeutic drug(s) to delay or reverse AD [19,20,21,22]. Thus, this hypothesis generating narrative review posits that sodium thiosulfate (STS) may act as a multi-targeting, multifactorial novel therapeutic to attenuate the development of LOAD.

Sodium thiosulfate (STS) has multiple industrial, nutritional, and medical uses. Industrial uses include its photographic fixative properties, water dechlorination, the removal of cyanide in mining gold, silver and lead, paper and pulp manufacturing, dyeing to decrease the intensity of colors, and cement manufacturing properties to name a few. Nutritional uses include STS addition to salt and alcohol to serve as a preservative. STS medical uses include the treatment of acne and tinea versicolor, cyanide and carbon monoxide intoxication, cisplatin oxidative-stress-induced ototoxicity, and calciphylaxis that associates with end-stage renal disease and dialysis [23,24,25,26,27,28,29,30,31,32].

STS (Na_2_S_2_O_3_) acts as a potent antioxidant reducing agent with its readily available and donatable two unpaired electrons that are capable of quenching the unpaired electrons of reactive oxygen species such as superoxide (O_2_^−^), hydrogen peroxide (H_2_O_2_) or hydroxyl group (−OH), and peroxinitrite (ONOO−) while it also is capable of being converted to the vasodilating gasotransmitter and neurotransmitter modulator hydrogen sulfide (H_2_S) and the antioxidant glutathione (GSH) (Figure 3) [24,30].

STS is an intermediate of sulfur metabolism and a known H_2_S donor in vivo through non-enzymatic and enzymatic mechanisms [31,33,34,35,36,37]. Enzymatic pathways include cystathionine beta-synthase (CBS), cystathionine gamma-lyase (CGL or CSE), and importantly the 3-mercaptopyruvate sulfurtransferase (3-MST). Non-enzymatic synthesis occurs through utilizing glucose, glutathione, inorganic and organic polysulfides, elemental sulfur, and the reverse trans-sulfuration pathway [35,38].

STS is known to be a clinically relevant source of hydrogen sulfide (H_2_S) via its ability to provide thiosulfate, which can lead to formation of H_2_S and polysulfides via rhodanese activity and/or the reverse trans-sulfuration pathway [35]. Notably, H_2_S is now considered to be the third gasotransmitter capable of instigating cerebral vasodilation similar to nitric oxide (NO) or carbon monoxide (CO) in addition to being a neurotransmitter modulator that is capable of generating sulfhydryl (−SH) thiol groups (Figure 4) [33,34].

Endogenous H_2_S acts as a gasotransmitter (capable of inducing cerebral vasodilation) and neuromodulator that exhibits neuroprotective effects by reducing oxidative stress, mitigating inflammation, and enhancing mitochondrial function [33,34]. H_2_S plays an important role in protecting neurons from damage in LOAD, stroke, and traumatic brain injury. Additionally, H_2_S has anti-apoptotic properties due to its antioxidant effects and improved mitochondrial function, along with its ability to modulate signaling pathways that regulate apoptosis such as survival factors including Bcl-2. Plus, H_2_S/STS contributes to anti-apoptotic signaling via the inhibition of JNK phosphorylation [33]. H_2_S can also regulate neuronal signaling pathways and contribute to cognitive function by modulating synaptic plasticity [33,39,40].

STS is an intermediate of sulfur metabolism and a known H_2_S donor in vivo through non-enzymatic and enzymatic mechanisms [31,34,35,36,37]. Enzymatic pathways include cystathionine beta-synthase (CBS), cystathionine gamma-lyase (CGL or CSE), and importantly the mitochondria and cytoplasmic 3-mercaptopyruvate sulfurtransferase (3-MST). Non-enzymatic synthesis occurs through utilizing glucose, glutathione, inorganic and organic polysulfides, elemental sulfur, and the reverse trans-sulfuration pathway [35,38]. Notably, STS does not provide for the direct release of H_2_S; however, it can provide its thiosulfate, which can lead to the formation of H_2_S via the mitochondrial rhodanese enzyme (also known as thiosulfate sulfurtransferase) activity as well as the reverse trans-sulfuration pathway [34].

Additionally, the impaired folate one-carbon metabolism (FOCM) pathway, which consists of interactive folate and methionine cycles to produce methyl groups for one-carbon donation, and the production of H_2_S via the trans-sulfuration pathway are important as they result in increased homocysteine (HCY) [3]. Hcy is a sulfur-containing amino acid that is not utilized in protein synthesis. However, Hcy serves as an important intermediate molecule in methionine metabolism as part of the FOCM pathway to provide methyl groups for one-carbon donation. Additionally, Hcy is located at a branch-point of the metabolic methionine and folate cycle pathways, which allows it to be either irreversibly degraded via the trans-sulfuration pathway to cysteine or remethylated to methionine. Importantly, as it enters the trans-sulfuration pathway, it is converted to cysteine, which can be converted enzymatically to H_2_S or glutathione (GSH) (Figure 5) [3].

Elevations of Hcy or hyperhomocysteinemia (HHCY) result in OxRS, which is associated with neurotoxicity [3,41,42,43]; whereas, H_2_S scavenges reactive species and protects neurons from OxRS [44,45,46,47]. Further, both the elevation of Hcy (HHCY) and the dysregulation of H_2_S have been detected in the plasma of individuals with LOAD [43,48], and the triad of HHCY, decreased folate, and dysregulated H_2_S have been detected in individuals with LOAD [3,43,48,49,50,51].

Individuals with LOAD have neurovascular unit (NVU) brain endothelial cell activation and dysfunction (BEC*act*/*dys*), with activation represented by a proinflammation and dysfunction manifesting as decreased NO bioavailability and blood–brain barrier dysfunction and disruption (BBB*dd*) with increased NVU permeability. Significantly, BEC*act*/*dys* and BBB***dd*** associate with impaired cognition and neurodegeneration [2,3,10]. Importantly, H_2_S is capable of inducing the production of NO via the activation the phosphatidylinositol 3-kinase (PI3K)/Akt signaling pathway with the subsequent increased phosphorylation of the eNOS enzyme at serine 1177 that results in increased NO production. Also, H_2_S is capable of promoting the S-sulfhydration and stabilization of the eNOS enzyme monomeric isoform to provide a more sustained production of NO [34,52]. Further, H_2_S neuroprotectant mechanisms provide an antioxidant, anti-inflammatory, and antiapoptotic effect in pathological situations [39]. Additionally, Abe et al. and Kimura et al. have established that H_2_S is a neuromodulator [53,54].

Indeed, STS as a H_2_S donor/mimetic provides neuromodulation by influencing behaviors of N-methyl-D-aspartate (NMDA) receptors and second messenger systems including intracellular Ca^2+^ concentration and intracellular cAMP levels in addition to neuroprotection provided by H_2_S antioxidant, anti-inflammatory, and antiapoptotic effects in pathological situations. Further, the sulfhydration of target proteins is an important mechanism underlying these effects [39,53].

Notably, STS acts as a hydrogen sulfide (H_2_S) donor/mimetic that also has multiple neuromodulating and neuroprotectant effects regarding the development and progression of LOAD (Figure 6) [31,32,33,34,52,53,54].

Each of the above four multifactorial effects in the neuroprotection and neuromodulation of LOAD will be discussed singularly in the following Section 2, Section 3, Section 4 and Section 5.

## 2. Antioxidant Effects of STS

The OxRS hypothesis (VI. in Box 1, Section 1) and the mitochondrial cascade hypothesis (VII. in Box 1, Section 1) intersection may not only be the earliest but also play the greatest effect on the development and progression of LOAD since it occurs prior to the development of cognitive impairment, synapse loss, and neurodegeneration [55,56]. Various groups have importantly implicated the role of dysfunctional or aberrant mitochondria (aMt) that become leaky and propagate further mitochondrial dysfunction such that OxRS instigates OxRS (Figure 7) [55,56,57].

In addition to the leakage of damaging ROS and RONSS causing OxRS, there are at least four negative consequences regarding the leakage of iron sulfur clusters (ISCs) from aMts that occur in LOAD. These include (1) increased OxRS (ROS and RONSS leakage) resulting in damage to lipids, proteins, and nucleic acids, which result in neuronal synapse and neuronal dysfunction that contribute to synaptic dysfunction and neurodegeneration; (2) the disruption of mitochondrial function resulting in decreased energy production. ISCs play a critical role in proper function of the electron transport chain including the production of ATP energy, and this decrease in ATP will lead to energy deficits within neurons with associated decline in synaptic and neuronal function with cognitive decline; (3) impaired cellular iron homeostasis resulting in iron excess and overload that contributes to increased OxRS in LOAD. Further, emerging evidence shows that excess iron with high redox activity is related to the deposition of amyloid plaques and the formation of neurofibrillary tangles, which suggests it may be one of the main causes of LOAD [58,59,60,61]; (4) altered protein function due to OxRS damage to proteins results in altered protein function, impaired transcription, translation, and repair processes and contributes to impaired cellular homeostasis and neurodegeneration [62,63,64,65].

OxRS includes reactive oxygen species (ROS), reactive nitrogen species (RNS), and reactive sulfur species (RSS) to form the reactive oxygen, nitrogen, and sulfur species (RONSS), which are responsible for the formation of the reactive species interactome (RSI). The activation of the RSI results in an early vicious cycle of RONSS instigating RONSS and inflammation and inflammation instigating RONSS in the development and progression of LOAD. These reactive species implicate the importance for the homeostatic role of the antioxidant defense system that becomes overwhelmed and contributes to the accumulation of RONSS [55]. Further, note that redox stress is listed first in the four mechanisms in the dementia quartet as presented in Figure 2, which promote the development and progression of LOAD.

There has been growing evidence over the past two decades that dysfunctional RSS of the RSI leads to some pathologies including LOAD [66]. RSS includes H_2_S, –SH (sulfhydryl groups), low molecular weight persulfides, and protein persulfides, as well as organic and inorganic polysulfides. Further, Iciek et al. have suggested that the modulation of RSS levels in the cell by using their precursors could be a potential and promising therapeutic tool in the treatment of LOAD [66].

The OxRS hypothesis is a significant factor in the development and progression of LOAD that revolves around the imbalance between the excessive production of RONSS and the brains’ inability to quench or neutralize them with its known reduction in antioxidants as compared to other tissues and organs [55,56,57]. OxRS contributes to the development and progression of LOAD due to an increase in mitochondria ROS, cellular membraneous NADPH oxidase (NOX), and xanthine oxidase and the reaction of advanced glycation endproducts (AGEs) and their receptor (RAGE) generation of ROS [67].

STS is a potent antioxidant with its two readily donatable electrons that are capable of quenching or neutralizing RONSS via its chain-breaking antioxidant effects on the unpaired electrons as illustrated in Figure 3, Figure 4, Figure 6 and Figure 7 [24,29,30].

The mechanisms for misfolded tau formation and the development of accumulated hyperphosphorylated tau to form neurofibrillary tangles (NFTs—filamentous aggregates of hyperphosphorylated tau) are complex. The anti-oxidant potential of STS and its donor capacity to form H_2_S may have a positive effect of preventing NFTs due to interfering with and scavenging OxRS free radicals. These free radicals trigger the activation of redox sensitive kinases such as glycogen synthase kinase 3β (GSK3β) and the development and overexpression of cyclin-dependent kinase-5 (CDk5); plus, other kinases responsible for the hyperphosphorylation of tau result in misfolding of tau and the formation of NFTs. Notably, STS is known to have anti-oxidant effects and could act in a manner to supplement the natural-occurring antioxidant enzymes of neurons, which include superoxide dismutase (SOD), catalase, and glutathione peroxidases that are known to be deficient in individuals with LOAD. Thus, STS may decrease ROS-induced increased activity of neuronal kinases responsible for the hyperphosphorylation of tau and decrease the misfolding of this protein, which is known to be responsible for the development of NFTs [1,8,11].

## 3. Anti-Inflammatory Effects of STS

The inflammation hypothesis in LOAD posits that chronic neuroinflammation plays an important, early, and critical role in the pathogenesis and progression of this debilitating neurologic disease. This hypothesis further suggests that the chronic activation of the brain’s immune system contributes to Aβ oligomers and plaque formation, tau pathology, and neuronal damage and loss [68,69,70,71,72,73,74,75,76,77,78,79,80,81,82,83,84,85,86]. There are two core features—pathologies of LOAD (core feature 1: extracellular neurotoxic Aβ oligomers and plaques; core feature 2: intracellular tau NFTs) that have helped to explain the development of LOAD. However, gaps have remained in the complete understanding of this complex chronic, progressive, and multifactorial disease. Over the past decade, a third core feature in the development and progression has emerged, which includes the neuroimmune system and chronic neuroinflammation [86]. Recently, our group was able to demonstrate, in lipopolysaccharide (LPS) (3 mg/kg)-induced CD-1 male rodent models of neuroinflammation at 28 h, the presence of BBB disruption. These models also demonstrated pronounced ultrastructural remodeling changes in brain endothelial cell(s) (BECs) of the neurovascular unit (NVU) including plasma membrane ruffling, increased numbers of extracellular microvesicles, small exosome formation, aberrant BEC mitochondria, and increased BEC transcytosis. Notably, while these remodeling changes were occurring, the tight and adherens junctions appeared to be unaltered. Also, aberrant pericytes were noted to be contracted with rounded nuclei and a loss of their elongated cytoplasmic processes, while multiple surveilling microglial cells were attracted to the NVU BECs. Interestingly, astrocyte detachment and separation were associated with the formation of enlarged perivascular spaces and reactive perivascular macrophages within the perivascular space. These previous ultrastructure remodeling changes were obtained from the frontal cortex in layer III [87].

Recent studies have demonstrated that the STS that is in equilibrium with H_2_S attenuates the neurotoxic effect of LPS-induced reactive microglia cells (rMGCs) and reactive astrocytes (rACs) in vitro glia–neuron co-cultures and protects mice against ischemic brain injury as well as LPS-induced acute lung injury [31,37,75,88,89].

Notably, Acero et al. were able to show that STS treatment was able to reduce the levels of the proinflammatory cytokine interleukin-1b (IL-1b), ionized calcium-binding adaptor molecule 1 (Iba-1), and the 18 kDa translocator protein (TSPO) in C57Bl6 mouse models of systemic LPS-induced neuroinflammation [89]. Further, Lee et al. were able to show that STS treatment (100 and 350 mg/kg) resulted in the reduced release of the proinflammatory cytokines [31]. Reductions in tumor necrosis alpha (TNFα), interleukin-6 (IL-6), and the attenuation of P38 mitogen-activated protein kinase (P38 MAPK) and nuclear factor kappa B (NFκB) proteins while increasing H_2_S and GSH were achieved in female C57BL/6J models [31]. Specifically, Lee et al. found that STS increased H_2_S and GSH expression in human microglia and astrocytes [31]. Specifically, Lee et al. have stated that STS may be a candidate for treating neurodegenerative disorders that have a prominent neuroinflammatory component [31]. Additionally, Marutani et al. were able to demonstrate that STS acted as a carrier molecule to allow H_2_S to have cytoprotective effect against neuronal ischemia and further established that thiosulfates exert antiapoptotic effects via the persulfidation of caspase-3 [37]. While STS is thought to not readily penetrate the intact NVU BBB, it is known that STS increases thiosulfate levels in the brain, choroid plexus, and cerebrospinal fluid [37,51,90].

Multiple evidence from laboratory and clinical studies has suggested that anti-inflammatory treatments could delay the development and progression of LOAD [37,51,69,88,89]. Indeed, neuroinflammation plays a critical role in the pathogenesis and progression of AD [69,91], and, in this section, the author has pointed to how the targeting of neuroinflammation with STS may provide an effective treatment strategy for the development and progression of LOAD by providing an anti-inflammatory approach and therapy before there is significant neuronal loss. When one places Aβ oligomers, Aβ plaques, and tau NFT centrally, it becomes obvious that there are both upstream and downstream proinflammatory mechanisms that are in play (Figure 8) [92].

While Figure 8 focuses primarily on the role of peripheral inflammation (upstream) and central neuroinflammation (downstream), it is important to also note that the vicious and self-perpetuating cycle between neuroinflammation and OxRS is also concurrently affecting aberrant remodeling changes within the brain resulting in neurodegeneration.

## 4. Chelation Effects of STS in Regard to Excess Calcium, Iron, and Copper

STS is a known chelator of cations such as the alkaline earth-metal calcium (Ca++) and the transition metals iron (Fe++) and copper (Cu++) [23,24,25,26,27,28,29,30,31,32]. Its role as a chelator of Ca++ ions is thought to be one of its primary effects on the positive outcomes in the clinical treatment of calciphylaxis [23,24,29,30,31]. In this section, the focus will be on STS multifactorial effects (as in Figure 6, Section 1) regarding its possible role in reducing the excess Ca++ overload in post-synaptic neurons and dendrites associated with glutamate excitotoxicity and neurodegeneration in LOAD. Also, the role of STS in the treatment of iron overload due to the development of the microvessel small vessel disease and cerebral microbleeds will be discussed [95].

### 4.1. Glutamate Excitotoxicity (GlutET) and Calcium (Ca^2+^) Excess/Overload

Glutamate is the most commonly engaged neurotransmitter in the CNS of mammals, and its actions contribute to mediating excitatory neurotransmission. Further, GlutET is known to be an important player in the development and progression of LOAD in regard to dendritic synapse dysfunction and loss and neurodegeneration [96,97]. This mechanism is initially due to excess glutamate that results in the increased opening of neuronal calcium channels, which results in neuronal calcium overload and neurodegeneration. Additionally, GlutET is associated with excess calcium accumulation within the mitochondria that serve as a calcium sink within the neuronal cytoplasm of dendrites and axons in LOAD. This mitochondria calcium overload results in excessive mitochondrial Ca^2+^ loading and Mt dysfunction with leaking and or loss that allows for the release of mtROS, Ca^2+^ escape from mitochondria, and Ca^2+^ overload, cytochrome C escape, and caspase signaling to result in apoptosis that associates with post-synaptic neurons with dendritic and/or axonal neurodegeneration (Figure 9) (Box 2) [96,97,98,99,100,101].

Box 2Glutamate (GL) excitotoxicity in LOAD due to Aβ oligomer toxicity to cradling astrocyte endfeet (ACef) and excitatory amino acid transporter (EAAT) receptor dysfunction. Note that the calcium excess and overload in addition to excessive mtROS in number 4 and increased ATP result in neurotoxicity with resulting neurodegeneration. Dysfunctional aberrant mitochondria may result in increased Ca++, increased mtROS, and early increase in ATP, and chronically decreased ATP due to mitochondrial pore formation and loss of mitochondria function is important in the mitochondria/ATP/ROS triangle [102]. ATP, adenosine triphosphate; Ca++, calcium; MtROS, mitochondrial derived reactive oxygen species; OxRS, oxidative redox stress; RSI, reactive species interactome.

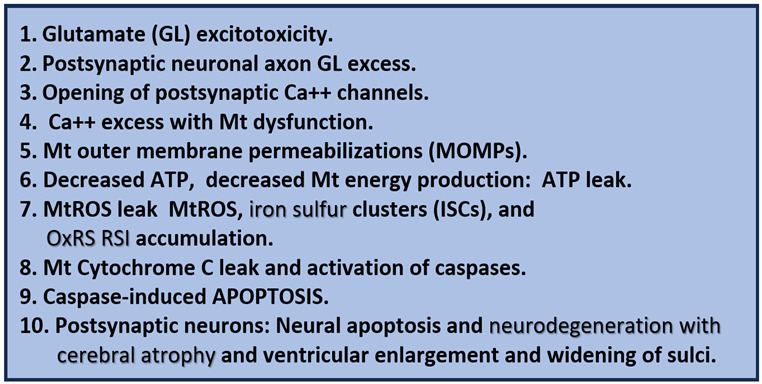



Importantly, microvessel SVD may be concurrently implicated with and contribute to the development and progression of LOAD [103]. Therefore, it is important to include the associated brain injury of microvessel SVD stroke (ischemic and/or hemorrhagic as in CMBs) and cerebral ischemia in the brain injury and the response to injury wound healing (BI:RTIWH) mechanisms since ischemia decreases ATP energy production, resulting in impaired excitatory amino acid transporter (EAAT) uptake of excess glutamate that is generated during ischemia (Figure 10) [104,105].

GlutET neurotransmission via the ligand-gated ionotropic glutamate receptor N-methyl-d-aspartate receptor (NMDAR) is critical for the synaptic plasticity and survival of neurons. The activation of synaptic NMDARs initiates synapse plasticity and stimulates cell survival, whereas the extrasynaptic NMDARs promote neuronal cell death and contribute to the development and progression of LOAD [105].

Thus, NMDAR-driven Ca^2+^ overload is not only neurotoxic but also an essential step in the GlutET cascade, resulting in neuronal dysfunction and neurodegeneration as in Figure 8 and Figure 9 and Box 2 [95,98,99,100,105,106]. This Ca^2+^ overload creates a potential treatment strategy with the employment of excessive calcium cation chelation by STS since STS is known to be an excellent chelator of calcium excess in the treatment of calciphylaxis [24,29,30,107]. Additionally, Zn^++^ is another cation in the GlutET cascade that plays an important role that may also be chelated by STS [108]. Previous studies have demonstrated that scavenging extracellular Ca^2+^ would decrease GlutET-induced neuronal degeneration, while the removal of other cations would not except for Zn^++^ [109,110]. This extracellular Ca^2+^ influx synergizes with the release of calcium from intracellular stores, such as the mitochondria and the endoplasmic reticulum, due to membrane damage caused by the disruption of ionic gradients [111].

Vascular ossification and calcification (VOC) in the brain associate with neurodegeneration and are a common finding in aging and in those individuals with LOAD by computed tomography (CT scans) and autopsy [112]. Notably, Subhash et al. have demonstrated that STS protects the brains of rat models with adenine-induced vascular calcification primarily by its antioxidant and calcium chelation effects [113]. Additionally, calcium dyshomeostasis and dysregulation via calcium excess and overload associate with GLET results in neurodegeneration and impaired cognition [114,115]. These findings in brain vascular calcification may certainly provide a mechanism for neuroprotection by STS via its antioxidative, calcium chelating, and anti-inflammatory multifactorial effects as previously discussed and presented in Figure 6 and Box 2.

### 4.2. Iron Overload Due to Microvessel Small Vessel Disease (mvSVD): Cerebral Microbleed(s) (CMBs) in LOAD

mvSVD and CMBs are known to be increased in LOAD, and this will increase neurotoxic hemoglobin, hemosiderin, and iron to increase neurodegeneration [95,116]. Iron accumulation in the brain exacerbates brain OxRS via the Fenton and Haber–Weiss reactions and leads to neuronal synaptic dysfunction and neurodegeneration [116,117,118]. mvSVD and CMBs are often clinically silent; however, these silent microbleeds may contribute to chronic iron excess and overload in individuals, especially as they age in addition to the prone to rupture CAA (considered an SVD) and intracerebral hemorrhages (ICHs) [119]. Further, iron overload associates with OxRS (Section 2 via the Fenton and Haber–Weiss reactions and neuroinflammation (Section 3), ferroptosis, and neurodegeneration [120]. Dixon et al. have proposed a fourth form of cell death mode that is due to iron overload that differs from apoptosis, necrosis, and autophagy termed ferroptosis [121]. Ferroptosis refers to an iron-dependent lipid peroxidation-induced cell death that depends on ROS production and iron excess, with severe lipid peroxidation [120]. Notably, Gleitze et al. have suggested that an iron–ferroptosis–Ca^2+^ mechanism may be involved in orchestrating the progression of impaired cognition that occurs in neurodegenerative diseases such as LOAD [122].

### 4.3. Copper (Cu) Dyshomeostasis and Cu Excess in LOAD

Cu plays an essential role as a redox catalyst of metalloproteins in electron-transfer (ET) sites in mitochondria, and, similar to iron, Cu is able to both donate and receive electrons [123]. Cu excess and toxicity are known to be present in LOAD; however, both Cu deficiency [124,125,126,127] and Cu excess [128,129,130,131,132,133,134] have been reported to associate with LOAD [123]. Indeed, Aβ associates with metal ions including Cu, Fe, and zinc. Some have even identified metal dyshomeostasis to be representative of a neurotoxic factor in individuals with LOAD [123,135].

Neurotoxic effects of metal dyshomeostasis (including Cu and Fe) frequently associate with impaired or reduced enzymatic activities, increased protein aggregation, and OxRS in the CNS, which allow for a cascade of events leading to cell death and neurodegeneration [135]. Specifically, the Cu–Aβ complex is known to catalyze the generation of reactive oxygen species that associates with OxRS and surrounding damage to cells and tissues [123].

Interestingly, in those individuals known to have LOAD, the disparity between those with excessive Cu and those with Cu deficiency may allow for a more personalized approach to treatment in regard to lowering Cu or correcting for its deficiency. While there exists a specific Cu chelator for Cu excess clioquinol [136], STS also has known Cu chelation properties and has been utilized in veterinary medicine for copper toxicity in sheep [137]; however, it may not be as effective as clioquinol or EDTA. Further, the findings in sheep may not be translatable to humans.

## 5. Neurovascular Effects of STS: Restoration of the Neurovascular Unit Brain Endothelial Cell Pro-Oxidative, Proinflammatory, and Proconstrictive State (Vascular Hypothesis)

STS is known to have antioxidant (Section 2), anti-inflammatory (Section 3), and anti-calcification–chelation (Section 4) properties in addition to its vasodilatory properties as discussed in this section [32].

The NVU is a complex functional and anatomical structure comprised of (1) neurons and interneurons, (2) glial cells such as microglia, astrocytes, and oligodendrocytes, (3) vascular mural cells including BECs, Pcs in capillaries, and VSMCs in arterioles, and (4) a BM (basal lamina) formed primarily by brain endothelial cells and pericytes that are both interspersed within the BM (Figure 11) [10,93,94,138,139].

True capillaries as in Figure 11 are unique in that the pia membrane (glia limitans) abruptly ends at the true capillary, and, therefore, true capillaries do not have perivascular space(s) (PVSs). Importantly, true capillaries are responsible for the uptake of nutrients, oxygen, and water and the efflux of metabolic waste [139,140,141], while post-capillary venules with their perivascular space(s) (PVSs) serve as a conduit for interstitial metabolic waste disposal via the PVS or glymphatic system. Additionally, the PVSs of the postcapillary venules allow for the transmigration of immune cells (innate and adaptive leukocytes) into the CNS parenchyma, as described by Owens et al. [142]. This strongly implicates the PVS in the two-step process of neuroinflammation, with step one consisting of circulating immune cells rolling, adhering, and transmigrating across the BECs of post-capillary venules and, step two, the progression of immune cells across the PVS and migration across the ACef basement membrane into the CNS parenchymal interstitium (Figure 12) [103,141,142].

RONSSs are known to be important signaling molecules in health. However, excessive RONSSs as manifested by OxRS are damaging to proteins, lipids, carbohydrates, and nucleic acids as presented in Section 2. OxRS acts as an injury that instigates the classic BI:RTIWH mechanism cascade as presented in Figure 10. These BI:RTIWH mechanisms result in neuroinflammation (acute and/or chronic) phase, granulation–proliferation phase with astrogliosis and angiogenesis, and a remodeling phase. Concurrently, this OxRS results in BEC*act*/*dys*, in which dysfunction manifests as decreased BEC-derived NO bioavailability via eNOS enzyme uncoupling. Homeostatic endothelial nitric oxide synthase enzyme (eNOS) coupling is dependent on the proper functioning of the essential tetrahydrobiopterin (BH_4_) cofactor [137,138]. The totally reduced form of BH_4_ is known to be an essential requisite cofactor to allow the eNOS to run and allow the synthesis of the vasodilatory gasotransmitter nitric oxide (NO) (Figure 13) [24,25,27,28,29,30,137,138,143,144,145].

Importantly, BH_4_ is known to be decreased in the brains and CSF of human individuals with LOAD [144,145].

The essential fully reduced BH_4_ cofactor is required to run the eNOS reaction, and if it becomes oxidized to BH_2_ via increased OxRS in LOAD, it will not allow the eNOS reaction to synthesize bioavailable NO due to eNOS uncoupling with decreased bioavailable NO. STS antioxidant properties allow it to restore oxidized BH_2_ to the requisite totally reduced BH_4_ cofactor to increase the production of bioavailable NO as in Figure 11. Also, STS actions as a H_2_S donor/mimetic allow it to stabilize the monomeric isoform of eNOS enzyme via S-sulfhydration to increase the production of bioavailable NO. The ability of STS to increase intracellular H_2_S allows for increased NO generation via increasing the activation of the PI3K/AKT pathway to promote eNOS phosphorylation and increase bioavailable NO via its ability to promote eNOS phosphorylation as previously discussed in the last three paragraphs of Section 1 [144,145].

Additionally, STS and H_2_S via their antioxidant and anti-inflammatory mechanisms will aid in promoting the restoration of normal BEC NO production and BBB function to result in an attenuation of the increased permeability associated with LOAD. Once STS and H_2_S have restored BEC*act*/*dys* and BBB***dd***, the enlarged perivascular spaces that associate with impaired clearance of Aβ oligomers, Aβ, and tau proteins will undergo improved clearance and result in a decrease in AB and tau with the attenuation of neurotoxicity and neurodegeneration [102,146].

## 6. Possible Intravenous Sodium Thiosulfate Treatment Paradigms in LOAD

Intravenous STS has been used in human individuals to prevent ototoxicity since 1985 via its known antioxidant effects against cisplatin with a dosage of 20 g per square meter, administered intravenously over a 15-min period, 6 h after the discontinuation of cisplatin [147]. Additionally, intravenous STS treatment paradigms for individuals with chronic-renal-failure-related hemodialysis and calciphylaxis have been previously described [23,24,25,26,27,28,29,30]. Importantly, in most studies, intravenous STS administration was provided as an add-on treatment to other standard treatment protocols for calciphylaxis. Most studies have supported the use of intravenous STS at a dosage of 25 g (two 12.5 g vials diluted in 100 cc of normal saline) during the last hour of hemodialysis, three times per week, and some suggest that 12.5 g per 100 cc of normal saline be used initially over a one-hour infusion as a test dose initially and, if tolerated, proceed to 25 g [23,24,148,149,150,151,152,153,154,155,156,157,158,159,160,161,162]. Additionally, STS has been used with peritoneal dialysis [146] and in pediatric patients (25 g/1.7 m^2^) [163]. There is considerable neuropathic pain that is associated with the ischemic skin and subcutaneous lesions of calciphylaxis that are magnified during hemodialysis. This pain is thought to be the result of subcutaneous and skin ischemia that associates with subcutaneous vascular and endoneurium calcification [24]. One of the qualities of STS multi-targeting treatments in these individuals with calciphylaxis was the rapid relief of this severe neuropathic pain, and this has been thought to be due to the restoration of the pathologic microvascular eNOS uncoupling, which associates with vascular ischemia and associated vascular and endoneurium calcification [24,30].

The duration of STS therapy depends on each individual patient; however, current thoughts are that intravenous STS should be used for at least two months beyond complete healing of the skin ulcerations in individuals with calciphylaxis [24,27,28,30]. Side effects of intravenous STS consist of nausea, abdominal cramping, vomiting and/or diarrhea if infused too rapidly (less than one hour). Bone density should be monitored if STS is used long term since STS was demonstrated to decrease bone strength in a rat model that prevented vascular calcification [164].

In contrast to humans, rodent models tolerate orally administered STS, and mice treated at a dose of 3 mg/mL in their drinking water for 6 weeks modulated cardiac dysfunction and the extracellular matrix remodeling due to arterial venous fistula, in part, by increasing ventricular H_2_S generation [165]. Also, intraperitoneal injections have been successfully utilized in rodent models at a dose of 2 g/kg STS at zero and 12 h after intratracheal lipopolysaccharide [88]. To date, STS has been administered by intravenous infusions in human individuals and both orally, intraperitoneal and intravenous, to preclinical models. In the future, other modes of administration such as the expanding use of nano-technology or intrathecal administration are potential novel modes of administration in addition to intravenous therapy.

Since intravenous STS has not been previously reported or approved for use in human individuals to treat LOAD, there will be many different approaches necessary such as intravenous dosage frequency and duration of treatment. However, one might initially follow the treatment regime previously described and utilized for the treatment of human individuals with calciphylaxis. Incidentally, intravenous STS could be administered not only in hospital outpatient departments (similar to the intravenous administration of the newer anti-amyloid monoclonal antibodies (MABs) treatments) but also in existing stand-alone infusion centers. Once the appropriate protocols were established, clinical trials could begin immediately since STS is already FDA approved even if it is being utilized as ‘off-label use’ similar to how it is used now globally for the treatment of calciphylaxis. The currently FDA-approved treatments for LOAD were introduced in Section 1. Currently, most interest is directed at anti-amyloid therapy. Monoclonal antibodies (mabs) include lecanemab, aducanumab, donanemab-agbt, and ponezumab, which are administered by intravenous infusion therapy similar to how STS may be used to treat LOAD per the previously described calciphylaxis treatment protocols.

In addition to the various treatment paradigms for intravenous STS treatment, it is appropriate to present the various medical uses, side effects, and future uses of STS (Figure 14).

## 7. Conclusions and Future Directions

This hypothesis generating this narrative review has provided a background outline for the use of sodium thiosulfate in the treatment of late-onset Alzheimer’s disease. Sodium thiosulfates’ multifactorial mechanistic roles include the possible protection, prevention, and delay of the debilitating symptoms associated with late-onset Alzheimer’s disease. Sodium thiosulfates’ multifactorial mechanistic roles include its protective effects on the pathologic roles of oxidative redox stress, inflammation/neuroinflammation, the chelation of excess calcium associated with glutamate excitotoxicity, and iron due to hemorrhages and cerebral microbleeds; its possible restorative neurovascular unit effects regarding the improvement of the brain endothelial cell and endothelial nitric oxide synthase enzyme; its essential cofactor tetrahydrobiopterin with restoration of nitric oxide production and bioavailability. The recoupling of the endothelial nitric oxide synthase enzyme allows brain endothelial cells to signal adjacent pericytes and/or vascular smooth muscle cells to relax and provide vasodilation to increase regional cerebral blood flow and may assist in abrogating chronic cerebral hypoperfusion to its regional surrounding synapses and neurons. Thus, the preventive and protective mechanisms of sodium thiosulfate may possibly contribute to a delay in the onset of late-onset Alzheimer’s disease with its debilitating morbidity and mortality (Figure 15).

One of the most vicious and self-perpetuating cycles in the development and progression of late-onset Alzheimer’s disease may be between oxidative redox stress and neuroinflammation wherein each are capable of instigating the other to result in neurodegeneration [166]. Brain endothelial cells, pericytes, microglia, and astrocytes are each sensitive to and capable of instigating and perpetuating this cycle once they are in a reactive or activated state that contribute to the development of neurodegeneration. Importantly, sodium thiosulfate and hydrogen sulfide are capable of breaking this vicious cycle via their antioxidant and anti-inflammatory effects as previously discussed in Section 2 and Section 3 and presented in Figure 6.

While this novel treatment paradigm with sodium thiosulfate is encouraging, there is a critical need for further research to further elucidate the mechanisms of action and to conduct rigorous clinical trials. Future studies should focus on optimizing treatment protocols and exploring their roles not only as a single therapy but also its use as an adjunct therapy in diverse populations. As we strive to develop effective interventions for Alzheimer’s disease, STS could represent a significant step forward in improving patient outcomes and quality of life. Continued exploration in this area of research in animal models and humans is not only warranted but essential, given the growing burden of the global aging baby boom generation (born between 1946 and 1964) and this age-related chronic neurodegenerative disease. Late-onset Alzheimer’s disease not only affects each individual’s morbidity and mortality but also affects the loving and caring that their care providers (usually family) offer and the cost to our society in providing care facilities and treatment.

Multi-target-directed ligands have recently been proposed as a new paradigm in the treatment of multifactorial diseases such as late-onset Alzheimer’s disease (LOAD) in addition to multi-targeted designed drugs [16,167]. Sodium thiosulfate may be best classified as a ligand, particularly in the context of its coordination with metals or other ions, rather than a pharmacophore, which refers to a specific structural feature of molecules that enables them to interact with biological targets in a drug-like manner.

In conclusion, sodium thiosulfate presents a novel multi-target and repurposed therapeutic strategy that may be a promising avenue for treating the multiple aberrant pathological mechanisms and structural remodeling changes associated with the development and progression of late-onset Alzheimer’s disease. Sodium thiosulfate’s potential neuromodulating and neuroprotective effects coupled with its ability to mitigate oxidative redox stress, neuroinflammation, the chelation of excessive calcium and iron accumulation due to glutamate excitotoxicity, cerebral microbleeds, and cerebral amyloid angiopathy, along with its possibility to restore brain endothelial cell and endothelial nitric oxide synthase coupling with increased nitric oxide, highlight its relevance in the treatment of late-onset Alzheimer’s disease as presented in Figure 1, Figure 2, Figure 4, Figure 6, Figure 11, Figure 12 and Figure 13. Indeed, sodium thiosulfate and hydrogen sulfide may serve as a ‘loadstar’–lodestar guiding point in the multi-targeted treatment approach of individuals with late-onset Alzheimer’s disease due to their targeting of multiple pathological mechanisms as in Figure 1 [92,168,169].

## Figures and Tables

**Figure 1 pharmaceuticals-17-01741-f001:**
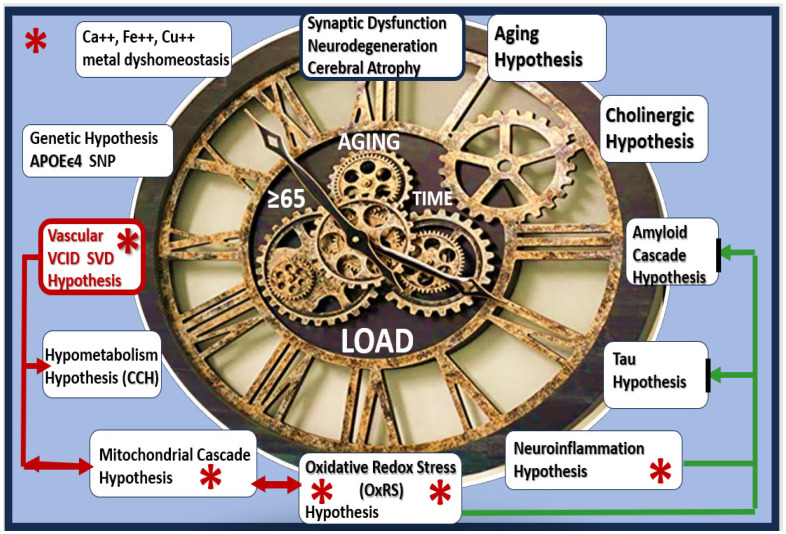
Late-onset Alzheimer’s disease (LOAD) is a multifactorial neurodegenerative disease with multiple hypotheses and multiple targets (1–12 o’clock on the clock face). Sodium thiosulfate (STS) (red asterisks) is known to target at least five of these 10 targets and, therefore, represents a novel multi-target treatment approach. Note that the positive effects of STS on the vascular hypothesis inclusive of the vascular contributions to impaired cognition and dementia (VCID) and cerebral small vessel disease (SVD) may also improve cerebral blood flow (CBF) and the chronic cerebral hypoperfusion (CCH) and in turn may have a positive effect on mitochondrial function with a decrease in oxidative redox stress (OxRS) (red lines with arrows) with consequent less neuroinflammation. Also, the aging and genetic hypotheses are non-modifiable. Aβ and tau misfolded protein accumulation (green lines with arrows). Aβ, amyloid beta; APOEϵ4, apolipoprotein E epsilon 4; Asterisks, indicate emphasis; Ca++, calcium; CCH, chronic cerebral hypoperfusion; Cu++, copper; Fe++, iron; SNP, single nucleotide polymorphism.

**Figure 2 pharmaceuticals-17-01741-f002:**
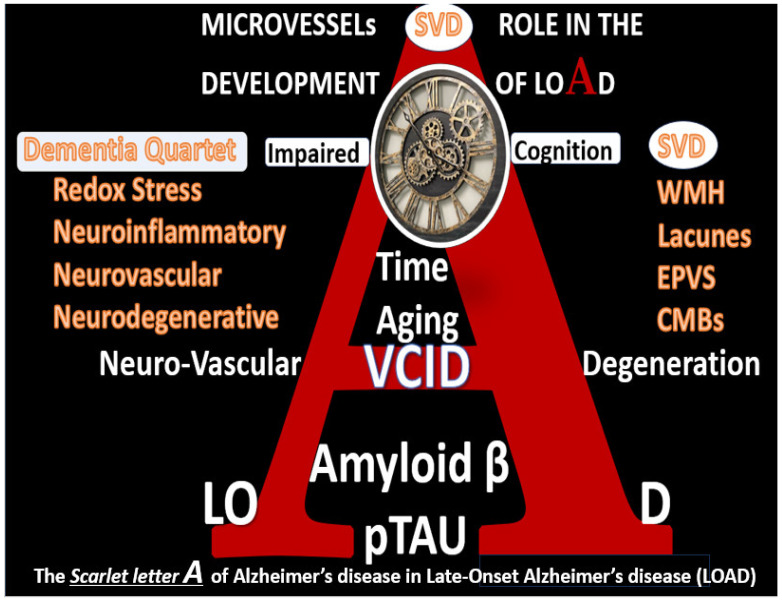
The scarlet letter A of Alzheimer’s disease in late-onset Alzheimer’s disease (LOAD). Note the dementia quartet mechanisms on the left-hand side of the figure and cerebral small vessel disease (SVD) are linked via LOAD and vascular contribution to cognitive impairment and dementia (VCID). Dementia quartet multifactorial mechanisms include (1) oxidative redox stress; (2) neuroinflammatory; (3) neurovascular; (4) neurodegenerative mechanisms in the development of LOAD. Image provided with permission by CC 4.0 [10]. CMBs, cerebral microbleeds; EPVS, enlarged perivascular spaces; WMH, white matter hyperintensities; VCID, vascular contributions to impaired cognition and dementia; underlining, indicates emphasis.

**Figure 3 pharmaceuticals-17-01741-f003:**
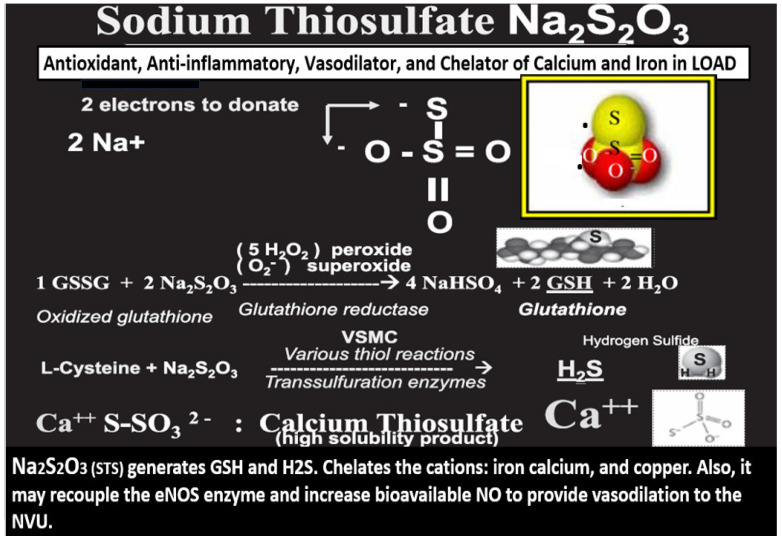
Sodium thiosulfate (STS) (Na_2_S_2_O_3_) acts as a chain-breaking antioxidant, and a chelator, and promotes vasodilation in late-onset Alzheimer’s disease (LOAD). Not only does STS act as a potent chain-breaking antioxidant due to its two unpaired electrons, but also STS is capable of generating the endogenous antioxidant, glutathione (GSH), and hydrogen sulfide (H_2_S). Further, STS is also capable of acting as a chelator of cations such as calcium (Ca++), copper (Cu++), and iron (Fe++). Modified image provided with permission by CC 4.0 [24,30]. eNOS, endothelial nitric oxide synthase; GSSG, oxidized GSH; H_2_O, water; H_2_S, hydrogen sulfide; NO, nitric oxide; NVU, neurovascular unit; VSMC, vascular smooth muscle cell.

**Figure 4 pharmaceuticals-17-01741-f004:**
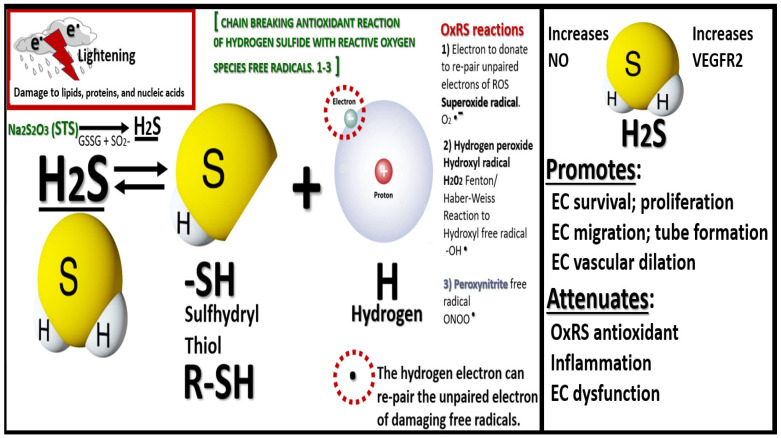
Sodium thiosulfate (STS/Na_2_S_2_O_3_ may be considered a hydrogen sulfide (H_2_S) donor, which is in equilibrium with the sulfhydryl thiol groups and hydrogen with its proton and unpaired free electron along with STS ability to act as a reducing agent to re-pair the unpaired electrons of damaging free radicals of superoxide, hydrogen peroxide and hydroxyl groups, and peroxinitrite labeled (1–3) in addition to undergoing S-sulfhydration reactions with various proteins. EC, endothelial cell–brain endothelial cell; GSSG, glutathione disulfide or oxidized glutathione (GSH); H, hydrogen; NO, nitric oxide; OxRS, oxidation redox stress; R, amino acid peptide/protein side chain; S, sulfur; VEGFR2, vascular endothelial growth factor receptor 2.

**Figure 5 pharmaceuticals-17-01741-f005:**
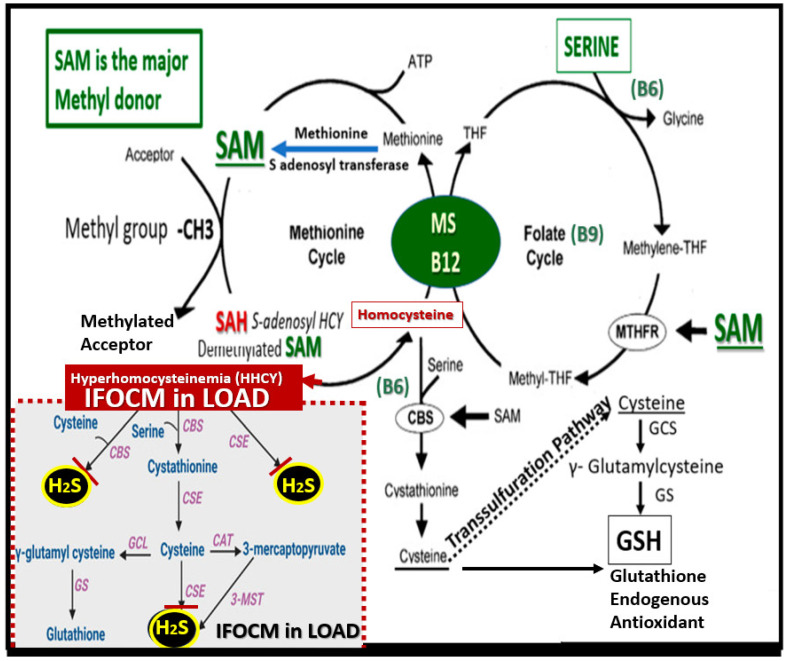
Impaired folate one-carbon metabolism (FOCM) in late-onset Alzheimer’s disease (LOAD) with hyperhomocystemia (HHCY) as a biomarker interferes with the normal function of the trans-sulfuration pathway in the production of hydrogen sulfide (H_2_S). This illustration demonstrates the normal methionine and folate cycles of normal FOCM and how the impaired FOCM (IFOCM) with HHCY that associates with LOAD impairs the trans-sulfuration pathway with decreased production of H_2_S (lower left panel outlined with red-dashed lines). The trans-sulfuration pathway yields H_2_S (antioxidant/anti-inflammatory, vasorelaxant angiogenic, and a neurotransmitter modulatory molecule, as well as producing the endogenous antioxidant glutathione (GSH). Importantly, HHCY would also impair the production of glutathione (GSH) in addition to depleting GSH due to increased oxidative redox stress. This modified image is provided with permission by CC4.0 [3]. B6, pyridoxal 5′-phosphate; B12, cobalamin; CAT, cysteine aminotransferase; CBS, cystathionine-beta-synthase; CSE, cystathionine gamma lyase (CGL); GCS, glutamate cysteine ligase (gamma-glutamylcysteine synthetase); GS, glutathione synthase; GSH, glutathione; 3MST, 3-mercaptopyruvate sulfurtransferase; MTHFR, methylenetetrahydrofolate; MS, methionine synthase; SAM, S-adenosylmethionine; THF, tetrahydrofolate.

**Figure 6 pharmaceuticals-17-01741-f006:**
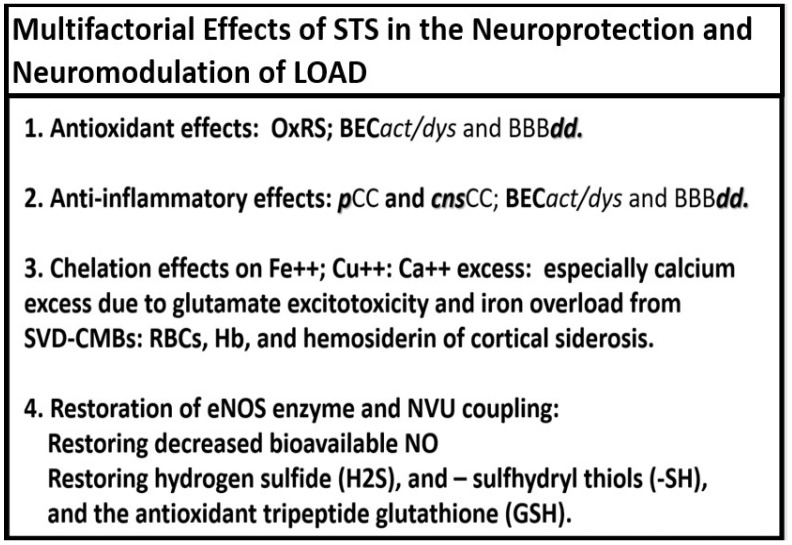
Multifactorial effects of sodium thiosulfate (STS) and hydrogen sulfide (STS-H_2_S) in the neuroprotection and neuromodulation of late-onset Alzheimer’s disease (LOAD). BBB***dd***, blood–brain barrier dysfunction and disruption; BEC*act*/*dys*, brain endothelial cell activation and dysfunction; Ca++, calcium; ***cns***CC, central nervous system cytokines/chemokines; Cu++, copper; eNOS, endothelial-derived nitric oxide synthase; Fe++, iron; Hb, hemoglobin; GSH, glutathione; NO, nitric oxide; NVU, neurovascular unit; OxRS, oxidative redox stress; ***p***CC, peripherally derived cytokines/chemokines; RBCs, red blood cells; –SH, sulfhydryl thiols; SVD–CMBs, small vessel disease and cerebral microbleeds.

**Figure 7 pharmaceuticals-17-01741-f007:**
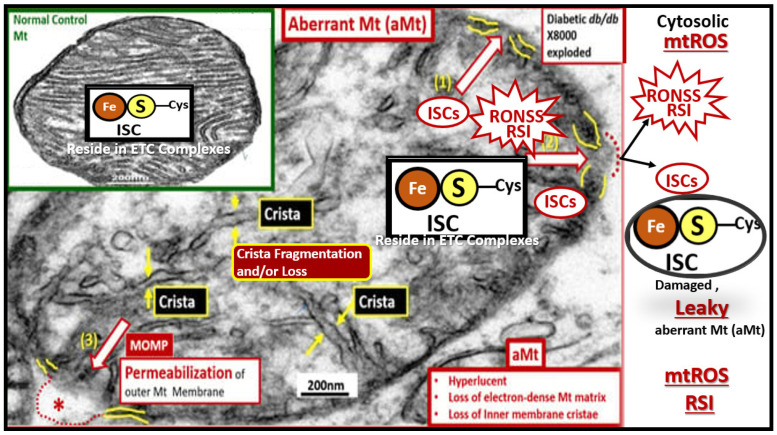
Aberrant mitochondria (aMt) are a source for multiple damaging reactive oxygen, nitrogen, and sulfur species (RONSS) that comprise the reactive species interactome (RSI). The RSI is a complex network of interactions between reactive species and their biological targets. aMt can be identified due to their hyperlucency, with loss of Mt matrix electron density with fragmentation and loss of cristae. Also note the iron (red), sulfur (yellow)–cysteine (Cys) clusters (ISCs) that are liberated from the aMt in addition to the RONSS as a result of aMt remodeling with the formation of mitochondrial outer membrane (yellow lines) permeabilization (MOMP) (red-dashed line and open arrows) with the creation of leaky Mt that leak ROS, RONSS, and ISCs. Note the insert upper left, which demonstrates a normal healthy Mt with intact ISCs that are maintained within the healthy Mt. Note the mitochondrial outer membrane permeabilization (MOMP): labeled (1) through (3), which allow the contents of the aMt to leak damaging RONSS and ISCs into the cytosol of neurons, allowing oxidative redox stress to damage lipids, proteins, and nucleic acids. Impaired mitophagy allow these leaky aMT to exist and continue to be leaky with damaging reactive species that eventually result in synaptic dysfunction and loss as well as neurodegeneration and impaired cognition. Thus, aMts are an important early and significant finding in the development and progression of LOAD. Revised image provided with permission by CC 4.0 [57]. This image is from the female obese diabetic *db*/*db* model at 20 weeks of age, and the insert is from a normal control model. Scale bars = 200 nm. Asterisk, indicates emphasis; ETC, electron transport chain.

**Figure 8 pharmaceuticals-17-01741-f008:**
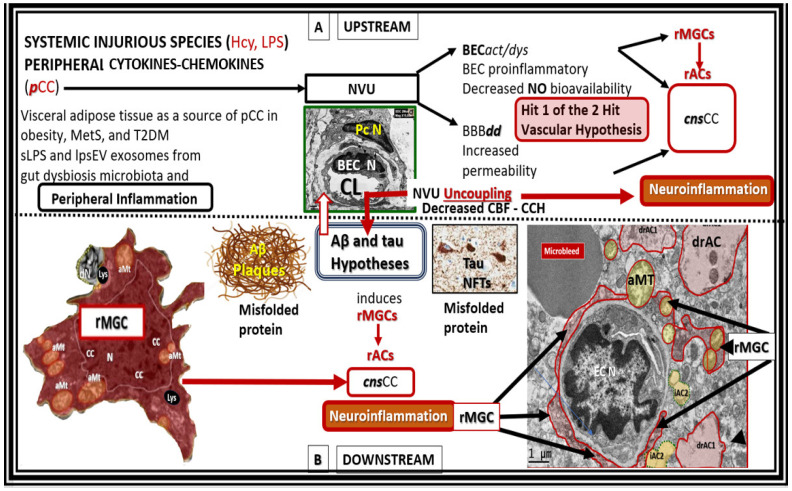
Upstream and downstream inflammatory effects on the development and progression of amyloid beta (Aβ) and tau in neurodegeneration. Upper panel (**A**) (upstream) depicts the effects of peripheral systemic injurious injuries such as homocysteine and lipopolysaccharides and peripheral cytokines and chemokines (***p***CC) and their effects on the neurovascular unit (NVU) that results in brain endothelial cell activation and dysfunction (BEC*act*/*dys*) in conjunction with blood–brain barrier dysfunction and disruption (BBB***dd***) providing Hit 1 of the 2-Hit vascular hypothesis. Lower panel (**B**) (below the horizontal dashed line is downstream) depicts the central importance of downstream Aβ and tau misfolded proteins that comprise Hit number 2 of the 2-Hit vascular hypothesis. Importantly, note the activation of microglia cell to reactive microglia cells (rMGCs) depicted on the far left and also the activation of astrocytes to reactive astrocyte (rACs) that are detached and retracted (drACs) from the NVU basement membrane on the far right of this image. Note that the rMGCs in the far-right image of the neurovascular unit appear to be invasive in addition to being attracted to the NVU that associate with detached and retracted drACs. Also, note that this image incorporates the vascular hypothesis in that it depicts hit-1 of Zlokovic’s 2-hit hypothesis in the upstream panel (**A**) [93,94]. aMT, aberrant mitochondria; BEC N, brain endothelial cell nucleus; CBF, cerebral blood flow; CCH, chronic cerebral hypoperfusion; cnsCC, central nervous system cytokines–chemokines; EC N, brain endothelial cell nucleus; MetS, metabolic syndrome; NFTs, neurofibrillary tangles; Pc N, pericyte nucleus; T2DM, type 2 diabetes mellitus.

**Figure 9 pharmaceuticals-17-01741-f009:**
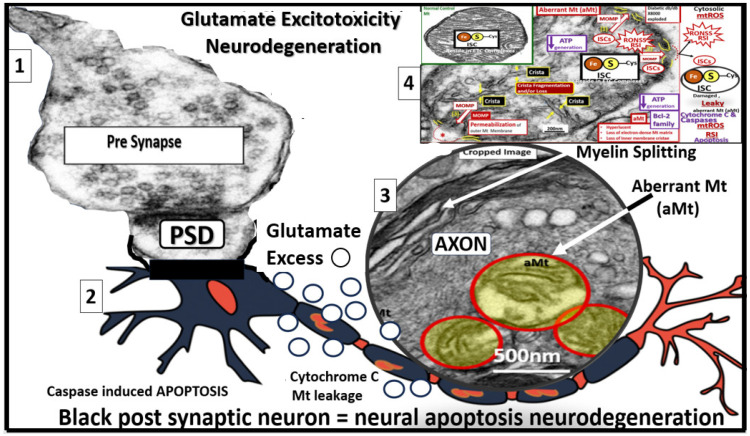
Glutamate excitotoxicity. Image 1 demonstrates a normal presynaptic neuron with multiple neurotransmitter vesicles. Image 2 depicts a blackened postsynaptic degenerative neuron due to glutamate excitotoxicity (open circles, accumulation of glutamate) promoting apoptotic neurodegeneration via excessive calcium and neurotoxicity. Image 3 depicts a transmission electron micrograph of an axon with three aberrant mitochondria (aMt) and myelin splitting. Image 4 depicts a compressed leaky aMt from previous Figure 7 that is responsible for excessive mtROS and calcium excess. Also, note the 10 steps involved with glutamate excitotoxicity to result in synaptic dysfunction and neurodegeneration as listed in Box 2. ATP, adenosine triphosphate; Ca++, calcium; GL, glutamate; Mt or mt, mitochondria; mtROS, mitochondrial reactive oxygen species.

**Figure 10 pharmaceuticals-17-01741-f010:**
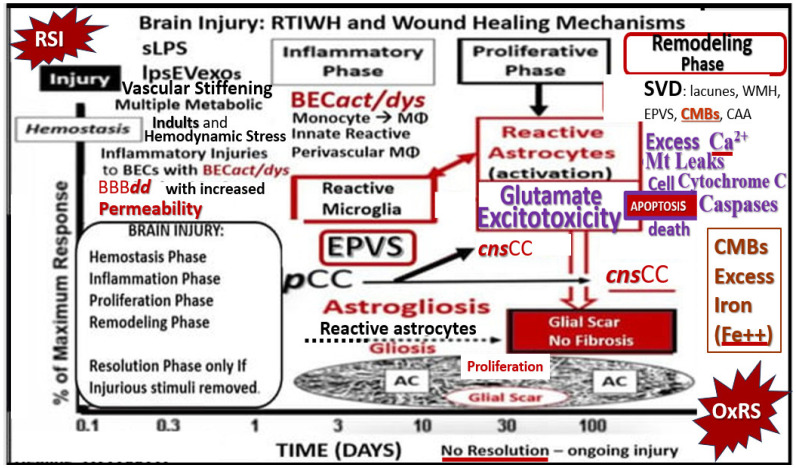
Late-onset Alzheimer’s disease (LOAD) and the brain injury: response to injury wound healing (BI:RTIWH) mechanism. While LOAD is considered to be a multifactorial disease, its initial injuries may be derived at the level of the brain endothelial cell of the neurovascular unit. However, the initial injury to neurons is thought to arise from the accumulation of misfolded proteins, which includes extracellular neurotoxic oligomers or plaques of amyloid beta (Aβ (1–42)) and intracellular tau neurofibrillary tangle(s) (NFTs) [11,12,13]. There are five basic phases in this BI:RTIWH mechanism that include 1. hemostasis; 2. inflammation; 3. proliferation; 4. remodeling; 5. resolution. Note that only phases 1–4 are shown and that the resolution phase is not included in this illustration because injurious stimuli are not removed in the development and progression of LOAD. Also, note that glutamate excitotoxicity and its subsequent effects leading to neuronal apoptosis and neurodegeneration are in purple color. Modified image provided with permission by CC 4.0 [104]. AC, astrocyte; *act*, activation; BECs, brain endothelial cells; CAA, cerebral amyloid angiopathy; Ca^2+^, calcium; CC, cytokines and chemokines; CMBs, cerebral microbleeds; CNS, central nervous system; ***cns***CC, central nervous system cytokines, chemokines; *dys*, dysfunction; EPVS, enlarged perivascular spaces; lpsEVexos, lipopolysaccharide extracellular vesicle exosomes; MetS, metabolic syndrome; MΦ, macrophage; Mt, mitochondria; ***p***CC, peripheral cytokines/chemokines; SVD, cerebral small vessel disease; T2DM, type 2 diabetes mellitus.

**Figure 11 pharmaceuticals-17-01741-f011:**
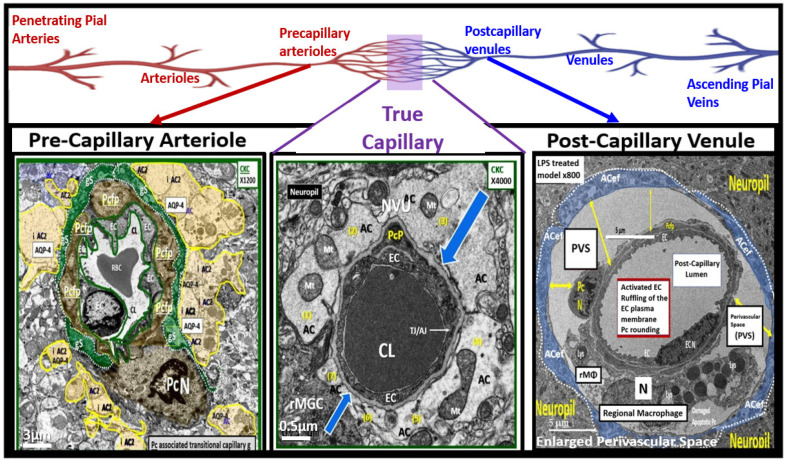
Representative cross-section transition electron microscopy (TEM) images of microvessels from various animal models from layer III in the frontal cortex at various magnifications to illustrate (1) precapillary arterioles (far left), true capillaries (center), and postcapillary venules (far right). These images are in contrast to those macrovessels that have a diameter of ≥5 μm with greater than 2 layers of vascular smooth muscle cell(s) (VSMCs) within their media. Note the perivascular space/nano-glymphatic space [pseudo-colored green] far left; the true capillary without a perivascular space (center) and an enlarged perivascular space (yellow double arrows for right). Magnification 3 μm, 0.5 μm, and 5 μm (far left, middle, and far right respectively). Note the upper elongated illustration of the macro- and microvessel pia vessels (arteries and arterioles in red color and vein and venules in blue color) and capillaries that correspond to the lower TEM images (arrows). Modified image provided with permission by CC 4.0 [10]. AC, astrocytes pseudo-colored gold in far-left and pseudo-colored blue far-right images. AQP-4, aquaporin 4; AC, perivascular astrocyte; AC1, AC2, astrocyte endfeet numbers 1 and 2 ACef, perivascular astrocyte endfeet; basement membrane (blue open arrows); CL, capillary lumen; EC, brain endothelial cell; gs, glymphatic space; lys, lysosome; Mt, mitochondria; N, nucleus; NVU, neurovascular unit; Pc, pericyte; PcN, pericyte nucleus; Pcp, pericyte endfeet processes; PVS, perivascular space; rMGC, interrogating or reactive microglia; rMΦ, reactive macrophage; TJ/AJ, tight junctions/adherens junctions.

**Figure 12 pharmaceuticals-17-01741-f012:**
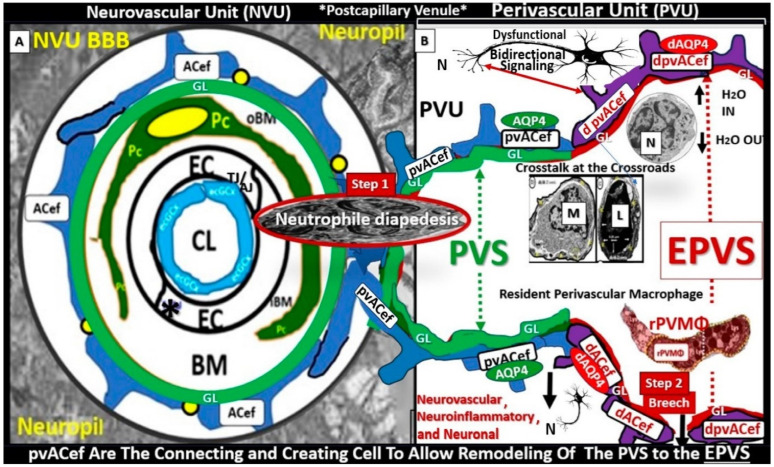
True capillary neurovascular unit (NVU) compared to the postcapillary venule perivascular unit (PVU). Protoplasmic perivascular astrocyte endfeet (pvACef) (pseudo-colored blue) within the true capillary (**A**) represent the creating and connecting ability of these cells, which allow remodeling of the normal PVU (**B**) and perivascular spaces (PVSs) to remodel into the pathologic enlarged perivascular space (EPV) that measures 1–3 mm on magnetic resonance imaging. (**A**) schematic illustrates the pseudo-colored blue true capillary neurovascular unit NVU (illustrating the transmission electron microscopic TEM true capillary image in Figure 11). Note that when the brain endothelial cells (BECs) become activated and NVU BBB disruption develops, due to BEC activation and dysfunction (BECact/dys) (due to multiple injurious species, which includes peripheral inflammation), there develops an increased permeability of fluids, peripheral cytokines and chemokines, and peripheral immune cells within the neutrophile (N) depicted herein penetrating the tight and adherens junction (TJ/AJ) paracellular spaces to enter the postcapillary venule along with monocytes (Ms) and lymphocytes (Ls) into the postcapillary venule PVS of the PVU B for step one of the two-step process of neuroinflammation. (**B**) depicts the postcapillary venule that contains the PVU. The PVU includes the normal PVS, which has the capability to remodel to the pathological EPVS. Notably, the proinflammatory leukocytes enter the PVS along with fluids, solutes, and cytokines/chemokines from an activated, disrupted, and leaky NVU in A. Further, note the pvACef (pseudo-colored blue) and its glia limitans (pseudo-colored cyan in the control NVU in (**A**) to the cyan color with exaggerated thickness for illustrative purposes in (**B**) that faces and adheres to the NVU BM extracellular matrix and faces the PVS PVU lumen that then becomes pseudo-colored red to denote the development of the EPVSs). Note that the pvACef have detached and separated, which allow for the creation of a perivascular space that transforms to an EPVS. The glia limitans becomes pseudo-colored red once the EPVSs have formed and then become breeched due to activation of matrix metalloproteinases and degradation of the proteins in the glia limitans. This remodeling allows neurotoxins and proinflammatory cells to leak into the interstitial spaces of the neuropil to mix with the ISF and result in neuroinflammation (step two) of the two-step process of neuroinflammation. Note that the dysfunctional pvACef AQP4 water channel is associated with the dysfunctional bidirectional signaling between the neuron (N) and the dysfunctional pvACef AQP4 water channel. Image provided by CC 4.0 [103]. AQP4 = aquaporin 4; asterisk = tight and adherens junction; BBB = blood–brain barrier; BM = both inner (i) and outer (o) basement membrane; dACef and dpvACef = dysfunctional astrocyte endfeet; EC = brain endothelial cell; ecGCx = endothelial glycocalyx; EVPS = enlarged perivascular space; fAQP4 = functional aquaporin 4; GL = glia limitans; H_2_O = water; L = lymphocyte; M = monocyte; N = neutrophile and neuron; Pc = pericyte; PVS = perivascular space; PVU = perivascular unit; rPVMΦ = resident perivascular macrophage; TJ/AJ = tight and adherens junctions.

**Figure 13 pharmaceuticals-17-01741-f013:**
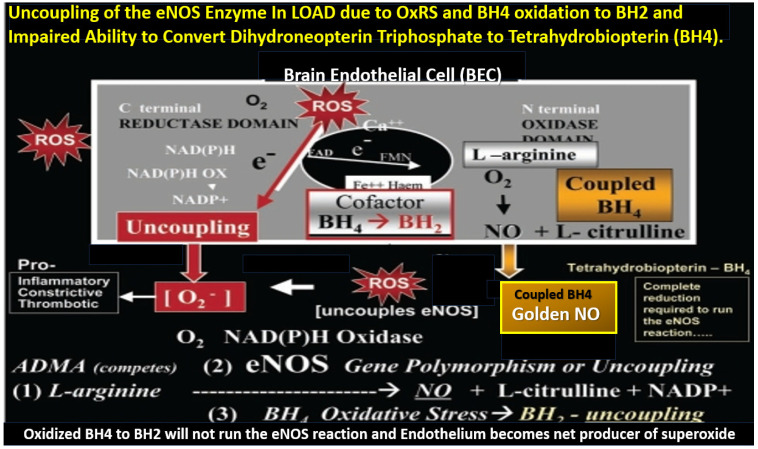
Endothelial nitric oxide synthase (eNOS) enzyme uncoupling results in the endothelium becoming a net producer of superoxide. eNOS enzyme uncoupling is depicted in this cartoon of the brain endothelium. Reactive oxygen species (ROS) and their oxidative effects on the requisite cofactor tetrahydrobiopterin (BH_4_) result in eNOS uncoupling is depicted. The excessive oxidation of BH_4_ results in the generation of BH_2_ that will not run the eNOS reaction to completion due to its incomplete reduction to BH_4_ to produce nitric oxide (NO). Instead, the reaction uncouples and shifts to the C terminal reductase domain, and oxygen reacts with the nicotine adenine dinucleotide phosphorus (NADPH)-reduced oxidase enzyme resulting in the generation of superoxide [O_2_^−^]. The oxidative redox stress (OxRS)-induced uncoupling of the eNOS reaction is known to result in a proinflammatory, proconstrictive, and prothrombotic endothelium, which contributes to endothelial activation and dysfunction (BEC*act*/*dys*). This modified image is provided with permission by CC 4.0 [24,30]. Ca++, calcium; FAD, flavin adenine dinucleotide; Fe++, iron; FMN, flavin mononucleotide; NADPH ox, nicotinamide adenine dinucleotide phosphate-reduced oxidase; O_2_, oxygen.

**Figure 14 pharmaceuticals-17-01741-f014:**
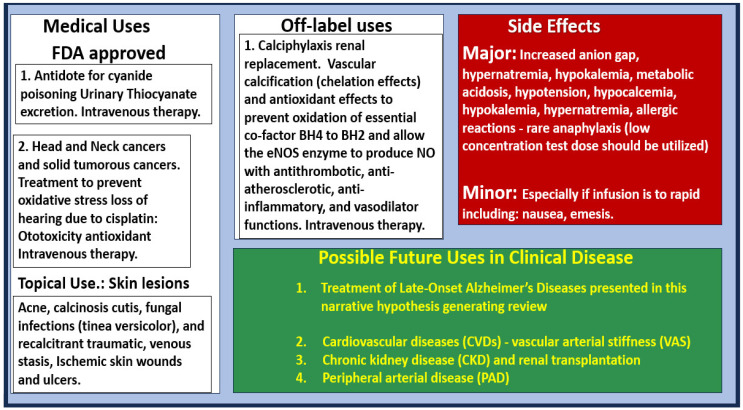
Medical uses, side effects, and possible future uses of sodium thiosulfate (STS) in clinical disease.

**Figure 15 pharmaceuticals-17-01741-f015:**
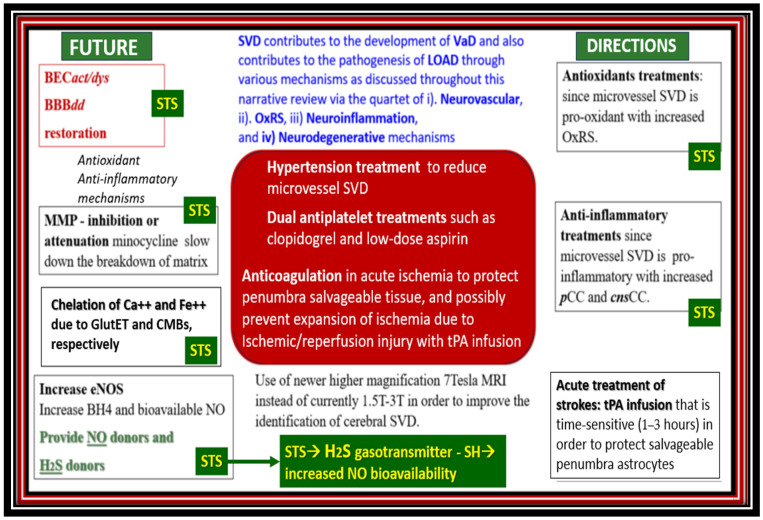
Future directions in the treatment of late-onset Alzheimer’s disease (LOAD) utilizing sodium thiosulfate (STS). This schematic illustrates the central role for the treatment of hypertension that is readily available for individuals with hypertension, proper anticoagulation when indicated, and the utilization of higher image magnification intensity of 7 tesla (7T). Note the surrounding seven boxes that are important to the development and progression of LOAD and how sodium thiosulfate (STS) is suggested as a treatment modality in 6 of these 7 boxed in underlying targets for STS. Also, note that dual antiplatelet therapy should also include cilostazol and low-dose aspirin. BBB***dd***, blood–brain barrier dysfunction and disruption; BEC*act*/*dys*, brain endothelial cell activation and dysfunction; BH_4_, tetrahydrobiopterin (essential cofactor for the eNOS enzyme); Ca++, calcium cation; CMBs, cerebral microbleeds; ***cns***CC, central nervous system cytokines/chemokines; e, electron; eNOS, endothelial nitric oxide synthase; Fe++, iron cation; GlutET, glutamate excitotoxicity; H_2_S, hydrogen sulfide; MMP, matrix metalloproteinases; MRI, magnetic resonance imaging; NO, brain endothelial-cell-derived nitric oxide; OxRS, oxidative redox stress; ***p***CC, peripheral cytokines/chemokines; SVD, small vessel disease; −SH, sulfhydryl group; tPA, tissue-type plasminogen activator; VaD, vascular dementia.

## Data Availability

The data and materials can be provided upon reasonable request.

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
