# Peer review of "Sodium Thiosulfate: An Innovative Multi-Target Repurposed Treatment Strategy for Late-Onset Alzheimer’s Disease"

_pharmaceuticals, 2024, doi:10.3390/ph17121741_

Round 1
Reviewer 1 Report
Comments and Suggestions for Authors
The authors presented a detailed review about Sodium Thiosulfate as a Multi-Target Strategy for Late-Onset Alzheimer’s Disease. This review focuses on a specific therapeutic agent, STS, and provides a comprehensive overview of its potential benefits for LOAD. Novelty is well appreciated. The article is very well written and organized. Wonderful figures, boxes and diagrams were provided. However, many revisions are required.
1- In the abstract, please illustrate potential side effects and safety concerns of STS especially in the context of long-term use.
2- In the abstract, please nominate new potential drug delivery strategies and the challenges associated with delivering STS to the brain, such as nanotechnology-based approaches. These novel approaches could enhance efficacy.
3- In line 321, write short explanation for tau pathology.
4- In figure 2 , the figure caption should include the full names for WMH, EPVS,CMBs.
5- In line 128, correct typo error in [ STS (Na2S2O32−)] it should be [ STS (Na2S2O3); 2 for Na should be subscript. Charge for S2O3 should be removed as it is neutral salt. Consider this rule in all the text such as line 141, 149, 168, 215, 230, and many others.
6- In figure 3 , the figure caption should include the full names for NVU.
7- The following related review article about multi-target strategies for Alzheimer's disease should be cited in the introduction
Ibrahim MM, Gabr MT. Multitarget therapeutic strategies for Alzheimer's disease. Neural Regen Res. 2019 Mar;14(3):437-440. doi: 10.4103/1673-5374.245463. PMID: 30539809; PMCID: PMC6334608.
8- In line 168, please explain briefly anti-apoptotic properties.
9- In line 188, remove folate one-carbon metabolism. Because it is previously stated in line 183; full name should be written for the abbreviation when first mentioned only.
10- In line 206, unify font style and size. Also, in line 670.
11- In line 231, write full name for NMDA.
12- In line 260, add a short definition for the reactive species interactome.
13- In line 479, consider correct superscript for ZN++
14- In line 517, consider writing zinc as Zn as other metals.
15- Correct typo error in line 654, result in a decrease of AB and tau with attenuation of neurotoxicity.
16- After line 701, the authors should discuss the available approaches for treatment of LOAD.
17- A diagram in section 6 about STS intravenous infusion uses and side effects is recommended.
18- Please discuss in future plan, novel potential drug delivery strategies and the challenges associated with delivering STS to the brain, such as nanotechnology-based approaches.
19- Conclusions should be written with full names for all abbreviations. Besides, it should be more precise.
20- Institutional Review Board Statement should be deleted. It is irrelvant for review article.
Author Response
Response to reviewer number 1 round (1):
First authors would like to thank reviewer number 1 for the precious time, effort and knowledge to review this manuscript. Authors responses will appear in blue color in this response and in the resubmitted manuscript.
Comments and Suggestions for Authors
The authors presented a detailed review about Sodium Thiosulfate as a Multi-Target Strategy for Late-Onset Alzheimer’s Disease. This review focuses on a specific therapeutic agent, STS, and provides a comprehensive overview of its potential benefits for LOAD. Novelty is well appreciated. The article is very well written and organized. Wonderful figures, boxes and diagrams were provided. However, many revisions are required.
Authors wish to thank reviewer number 1 for these kind comments.
1-In the abstract, please illustrate potential side effects and safety concerns of STS especially in the context of long-term use. Authors thank reviewer for this suggestion please see lines (26-28) as follows:
“Side effects have been minimal with reports of metabolic acidosis and increased anion gap, as with any drug treatment the possibility of allergic reactions, possible long-term osteoporosis from animal studies to date, and minor side-effects of nausea, headache and rhinorrhea if infused too rapidly.”
2- In the abstract, please nominate new potential drug delivery strategies and the challenges associated with delivering STS to the brain, such as nanotechnology-based approaches. These novel approaches could enhance efficacy. Please see lines (32-33) as follows: “Novel strategies such as the future use of nano-technology may be helpful in allowing an increased entry of STS into the brain.”
3- In line 321, write short explanation for tau pathology Authors wish to thank the reviewer number 1 for this important recommendation. Authors have now placed the following in the revised manuscript lines (325-338). “The mechanisms for misfolded tau formation and the development of accumulated hyperphosphorylated tau to form neurofibrillary tangles (NFTs-filamentous aggregates of hyperphosphorylated tau) are complex. The anti-oxidant potential of STS and its donor capacity to form H2S may have a positive effect of preventing NFTs due to interfering with and scavenging OxRS free radicals. These free radicals trigger the activation of redox sensitive kinases such as glycogen synthase kinase 3β (GSK3β) and the development and overexpression of cyclin-dependent kinase-5 (CDk5) plus other kinases responsible for the hyperphosphorylation of tau result in misfolding of tau and the formation of NFTs. Notably, STS is known to have anti-oxidant effects and could act in a manner to supplement the natural-occurring antioxidant enzymes of neurons which include superoxide dismutase (SOD), catalase, and glutathione peroxidases that are known to be deficient in individuals with LOAD. Thus, STS may decrease ROS-induced increased activity of neuronal kinases responsible for hyperphosphorylation of tau and decrease misfolding of this protein, which is known to be responsible for the development of NFTs [1, 8, 11].”
4- In figure 2 , the figure caption should include the full names for WMH, EPVS,CMBs. The full names of these abbreviations in the figure have been added to the legend for this figure. Please see lines (110-112) as follows: “CMBs, cerebral microbleeds; EPVS, enlarged perivascular spaces; WMH, white matter hyperintensities; VCID, vascular contributions to impaired cognition and dementia.”
5- In line 128, correct typo error in [ STS (Na2S2O32−)] it should be [ STS (Na2S2O3); 2 for Na should be subscript. Charge for S2O3 should be removed as it is neutral salt. Consider this rule in all the text such as line 141, 149, 168, 215, 230, and many others. Authors thank reviewers for this kind oversight recommenda tion. STS now appears in blue lettering as Na2S2O3
6- In figure 3 , the figure caption should include the full names for NVU. Authors have now added the abbreviations to the figure legend for figure 3 as follows in lines (148-149): “GSSG, oxidized GSH; H2O, water; H2S, hydrogen sulfide; NO, nitric oxide; NVU, neurovascular unit; VSMC, vascular smooth muscle cell.”
7- The following related review article about multi-target strategies for Alzheimer's disease should be cited in the text Ibrahim MM, Gabr MT. Multitarget therapeutic strategies for Alzheimer's disease. Neural Regen Res. 2019 Mar;14(3):437-440. doi: 10.4103/1673-5374.245463. PMID: 30539809; PMCID: PMC6334608. Authors have added this reference to the text… [168, 169] line 759 and appears in reference section as 169. Ibrahim MM, Gabr MT. Multitarget therapeutic strategies for Alzheimer's disease. Neural Regen Res. 2019 Mar;14(3):437-440. doi: 10.4103/1673-5374.245463 in lines 1202-1203.
8- In line 168, please explain briefly anti-apoptotic properties. This is a great recommendation and authors thank reviewer number 1. Authors have now placed a brief explanation as to how H2S/STS provides anti-apoptotic properties as follows: “Additionally, H2S has anti-apoptotic properties (not shown in figure 3.) due to its antioxidant effects and improved mitochondrial function, along with its ability to modulate signaling pathways that regulate apoptosis such as survival factors including Bcl-2. Plus, H2S/STS contributes to anti-apoptotic signaling via inhibition of JNK phosphorylation [33].” Lines (175-179).
9- In line 188, remove folate one-carbon metabolism. Because it is previously stated in line 183; full name should be written for the abbreviation when first mentioned only. Authors have made this change see line 192.
10- In line 206, unify font style and size. Also, in line 670. Authors have made these corrections throughout the revised manuscript.
11- In line 231, write full name for NMDA. This has been corrected as follows “behaviors of N-methyl-D-aspartate (NMDA) receptors and second messenger systems…”
12- In line 260, add a short definition for the reactive species interactome. Authors have now placed a brief definition of the reactive species interactome in lines (266-268) as follows: “Aberrant mitochondria (aMt) are a source for multiple damaging reactive oxygen, nitrogen, and sulfur species (RONSS) that comprise the reactive species interactome (RSI). The RSI is a complex network of interactions between reactive species and their biological targets” 13- In line 479, consider correct superscript for ZN++ Authors have now made these changes as follows:
“Additionally, Zn++ is another cation in the GlutET cascade that plays an important role that may also be chelated by STS [108]. Previous studies have demonstrated that scavenging extracellular Ca2+ would decrease GlutET induced neuronal degeneration, while the removal of other cations would not except for Zn++ [109, 110].”
14- In line 517, consider writing zinc as Zn as other metals. This has now been accomplished see 13. previous recommendation.
15- Correct typo error in line 654, result in a decrease of AB and tau with attenuation of neurotoxicity. Authors have made the following changes: “… and result in a decrease of AB and tau with attenuation of neurotoxicity and neurodegeneration [102, 146].”
16- After line 701, the authors should discuss the available approaches for treatment of LOAD. Authors have previously described the FDA-approved treatments for LOAD in the introduction in lines (74-83) as follows; “Indeed, these multiple hypotheses as presented in box 1 certainly support the concept that LOAD is a multifactorial, heterogenous disease that presents a dilemma in treatment when that treatment is for just one of these hypotheses. Multiple food and drug administration (FDA)-approved treatments over the years include 1). Cholinesterase inhibitors such as donepezil and galantamine; 2). Glutamate inhibitors such as memantine; 3). Anti-amyloid therapy - monoclonal antibodies (mabs) such as lecanemab, aducanumab, donanemab-agbt, and ponezumab have been approved but are still being clinically studied in various trials; 4). Various antipsychotics to modify impaired behavior and psychological symptoms such as brexpiprazole and newer insomnia medications including suvorexant [14, 15].”
Even though this was introduced earlier, authors decided to follow reviewers suggestions and added the following: “The currently FDA-approved treatments for LOAD were introduced in Section 1. (Introduction) and currently, most interest is directed at anti-amyloid therapy. Monoclonal antibodies (mabs) include lecanemab, aducanumab, donanemab-agbt, and ponezumab that are administered by intravenous infusion therapy similar to how STS may be used to treat LOAD per the previously described calciphylaxis treatment protocol.” Lines (707-712).”
17- A diagram in section 6 about STS intravenous infusion uses and side effects is recommended. Authors have created a new figure 14 to place the known uses and side effects that have been commonly observed with the use of sodium thiosulfate. See lines (734-741). As follows: “In addition to the various treatment paradigms for intravenous STS treatment, it is appropriate to present the various medical uses, side effects, and future uses of STS (Fig. 14).
Figure 14. Medical uses, side effects, and possible future uses of sodium thiosulfate (STS) in Clinical disease.”
18- Please discuss in future plan, novel potential drug delivery strategies and the challenges associated with delivering STS to the brain, such as nanotechnology-based approaches. Authors have added this to the abstract as per reviewer’s recommendations as per recommended in (2-). Also see lines (714-718) as follows: “intratracheal lipopolysaccharide [88]….. To date STS has been administered by intravenous infusions in human individuals and both orally, intraperitoneal and intravenous to preclinical models. In the future, other modes of administration such as the expanding use of nanotechnology or intrathecal administration are potential novel modes of administration in addition to intravenous therapy.”
19- Conclusions should be written with full names for all abbreviations. Besides, it should be more precise. Authors have now spelled out all of the abbreviations in the conclusion section lines (716-790).
20- Institutional Review Board Statement should be deleted. It is irrelvant for review article. Authors response: During the pre-check review by the editorial staff they wanted authors to add dates and that was done. So will leave in for now until the final proofing check should the editors and reviewers decide if manuscript is worthy of acceptance.
To reviewer number 1: please also see the responses to reviewer number 2.
Sincerely with gratitude,
Melvin R Hayden
Submitting author.
Response to reviewer number 1 round (1):
First authors would like to thank reviewer number 1 for the precious time, effort and knowledge to review this manuscript. Authors responses will appear in blue color in this response and in the resubmitted manuscript.
Comments and Suggestions for Authors
The authors presented a detailed review about Sodium Thiosulfate as a Multi-Target Strategy for Late-Onset Alzheimer’s Disease. This review focuses on a specific therapeutic agent, STS, and provides a comprehensive overview of its potential benefits for LOAD. Novelty is well appreciated. The article is very well written and organized. Wonderful figures, boxes and diagrams were provided. However, many revisions are required.
Authors wish to thank reviewer number 1 for these kind comments.
1-In the abstract, please illustrate potential side effects and safety concerns of STS especially in the context of long-term use. Authors thank reviewer for this suggestion please see lines (26-28) as follows:
“Side effects have been minimal with reports of metabolic acidosis and increased anion gap, as with any drug treatment the possibility of allergic reactions, possible long-term osteoporosis from animal studies to date, and minor side-effects of nausea, headache and rhinorrhea if infused too rapidly.”
2- In the abstract, please nominate new potential drug delivery strategies and the challenges associated with delivering STS to the brain, such as nanotechnology-based approaches. These novel approaches could enhance efficacy. Please see lines (32-33) as follows: “Novel strategies such as the future use of nano-technology may be helpful in allowing an increased entry of STS into the brain.”
3- In line 321, write short explanation for tau pathology Authors wish to thank the reviewer number 1 for this important recommendation. Authors have now placed the following in the revised manuscript lines (325-338). “The mechanisms for misfolded tau formation and the development of accumulated hyperphosphorylated tau to form neurofibrillary tangles (NFTs-filamentous aggregates of hyperphosphorylated tau) are complex. The anti-oxidant potential of STS and its donor capacity to form H2S may have a positive effect of preventing NFTs due to interfering with and scavenging OxRS free radicals. These free radicals trigger the activation of redox sensitive kinases such as glycogen synthase kinase 3β (GSK3β) and the development and overexpression of cyclin-dependent kinase-5 (CDk5) plus other kinases responsible for the hyperphosphorylation of tau result in misfolding of tau and the formation of NFTs. Notably, STS is known to have anti-oxidant effects and could act in a manner to supplement the natural-occurring antioxidant enzymes of neurons which include superoxide dismutase (SOD), catalase, and glutathione peroxidases that are known to be deficient in individuals with LOAD. Thus, STS may decrease ROS-induced increased activity of neuronal kinases responsible for hyperphosphorylation of tau and decrease misfolding of this protein, which is known to be responsible for the development of NFTs [1, 8, 11].”
4- In figure 2 , the figure caption should include the full names for WMH, EPVS,CMBs. The full names of these abbreviations in the figure have been added to the legend for this figure. Please see lines (110-112) as follows: “CMBs, cerebral microbleeds; EPVS, enlarged perivascular spaces; WMH, white matter hyperintensities; VCID, vascular contributions to impaired cognition and dementia.”
5- In line 128, correct typo error in [ STS (Na2S2O32−)] it should be [ STS (Na2S2O3); 2 for Na should be subscript. Charge for S2O3 should be removed as it is neutral salt. Consider this rule in all the text such as line 141, 149, 168, 215, 230, and many others. Authors thank reviewers for this kind oversight recommenda tion. STS now appears in blue lettering as Na2S2O3
6- In figure 3 , the figure caption should include the full names for NVU. Authors have now added the abbreviations to the figure legend for figure 3 as follows in lines (148-149): “GSSG, oxidized GSH; H2O, water; H2S, hydrogen sulfide; NO, nitric oxide; NVU, neurovascular unit; VSMC, vascular smooth muscle cell.”
7- The following related review article about multi-target strategies for Alzheimer's disease should be cited in the text Ibrahim MM, Gabr MT. Multitarget therapeutic strategies for Alzheimer's disease. Neural Regen Res. 2019 Mar;14(3):437-440. doi: 10.4103/1673-5374.245463. PMID: 30539809; PMCID: PMC6334608. Authors have added this reference to the text… [168, 169] line 759 and appears in reference section as 169. Ibrahim MM, Gabr MT. Multitarget therapeutic strategies for Alzheimer's disease. Neural Regen Res. 2019 Mar;14(3):437-440. doi: 10.4103/1673-5374.245463 in lines 1202-1203.
8- In line 168, please explain briefly anti-apoptotic properties. This is a great recommendation and authors thank reviewer number 1. Authors have now placed a brief explanation as to how H2S/STS provides anti-apoptotic properties as follows: “Additionally, H2S has anti-apoptotic properties (not shown in figure 3.) due to its antioxidant effects and improved mitochondrial function, along with its ability to modulate signaling pathways that regulate apoptosis such as survival factors including Bcl-2. Plus, H2S/STS contributes to anti-apoptotic signaling via inhibition of JNK phosphorylation [33].” Lines (175-179).
9- In line 188, remove folate one-carbon metabolism. Because it is previously stated in line 183; full name should be written for the abbreviation when first mentioned only. Authors have made this change see line 192.
10- In line 206, unify font style and size. Also, in line 670. Authors have made these corrections throughout the revised manuscript.
11- In line 231, write full name for NMDA. This has been corrected as follows “behaviors of N-methyl-D-aspartate (NMDA) receptors and second messenger systems…”
12- In line 260, add a short definition for the reactive species interactome. Authors have now placed a brief definition of the reactive species interactome in lines (266-268) as follows: “Aberrant mitochondria (aMt) are a source for multiple damaging reactive oxygen, nitrogen, and sulfur species (RONSS) that comprise the reactive species interactome (RSI). The RSI is a complex network of interactions between reactive species and their biological targets” 13- In line 479, consider correct superscript for ZN++ Authors have now made these changes as follows:
“Additionally, Zn++ is another cation in the GlutET cascade that plays an important role that may also be chelated by STS [108]. Previous studies have demonstrated that scavenging extracellular Ca2+ would decrease GlutET induced neuronal degeneration, while the removal of other cations would not except for Zn++ [109, 110].”
14- In line 517, consider writing zinc as Zn as other metals. This has now been accomplished see 13. previous recommendation.
15- Correct typo error in line 654, result in a decrease of AB and tau with attenuation of neurotoxicity. Authors have made the following changes: “… and result in a decrease of AB and tau with attenuation of neurotoxicity and neurodegeneration [102, 146].”
16- After line 701, the authors should discuss the available approaches for treatment of LOAD. Authors have previously described the FDA-approved treatments for LOAD in the introduction in lines (74-83) as follows; “Indeed, these multiple hypotheses as presented in box 1 certainly support the concept that LOAD is a multifactorial, heterogenous disease that presents a dilemma in treatment when that treatment is for just one of these hypotheses. Multiple food and drug administration (FDA)-approved treatments over the years include 1). Cholinesterase inhibitors such as donepezil and galantamine; 2). Glutamate inhibitors such as memantine; 3). Anti-amyloid therapy - monoclonal antibodies (mabs) such as lecanemab, aducanumab, donanemab-agbt, and ponezumab have been approved but are still being clinically studied in various trials; 4). Various antipsychotics to modify impaired behavior and psychological symptoms such as brexpiprazole and newer insomnia medications including suvorexant [14, 15].”
Even though this was introduced earlier, authors decided to follow reviewers suggestions and added the following: “The currently FDA-approved treatments for LOAD were introduced in Section 1. (Introduction) and currently, most interest is directed at anti-amyloid therapy. Monoclonal antibodies (mabs) include lecanemab, aducanumab, donanemab-agbt, and ponezumab that are administered by intravenous infusion therapy similar to how STS may be used to treat LOAD per the previously described calciphylaxis treatment protocol.” Lines (707-712).”
17- A diagram in section 6 about STS intravenous infusion uses and side effects is recommended. Authors have created a new figure 14 to place the known uses and side effects that have been commonly observed with the use of sodium thiosulfate. See lines (734-741). As follows: “In addition to the various treatment paradigms for intravenous STS treatment, it is appropriate to present the various medical uses, side effects, and future uses of STS (Fig. 14).
Figure 14. Medical uses, side effects, and possible future uses of sodium thiosulfate (STS) in Clinical disease.”
18- Please discuss in future plan, novel potential drug delivery strategies and the challenges associated with delivering STS to the brain, such as nanotechnology-based approaches. Authors have added this to the abstract as per reviewer’s recommendations as per recommended in (2-). Also see lines (714-718) as follows: “intratracheal lipopolysaccharide [88]….. To date STS has been administered by intravenous infusions in human individuals and both orally, intraperitoneal and intravenous to preclinical models. In the future, other modes of administration such as the expanding use of nanotechnology or intrathecal administration are potential novel modes of administration in addition to intravenous therapy.”
19- Conclusions should be written with full names for all abbreviations. Besides, it should be more precise. Authors have now spelled out all of the abbreviations in the conclusion section lines (716-790).
20- Institutional Review Board Statement should be deleted. It is irrelvant for review article. Authors response: During the pre-check review by the editorial staff they wanted authors to add dates and that was done. So will leave in for now until the final proofing check should the editors and reviewers decide if manuscript is worthy of acceptance.
To reviewer number 1: please also see the responses to reviewer number 2.
Sincerely with gratitude,
Melvin R Hayden
Submitting author.

Reviewer 2 Report
Comments and Suggestions for Authors§ What could be the possible formulation approaches to enhance the BBB permeability of sodium thiosulfate? Please include them.
§ Why are there too many keywords? The number can be limited by omitting the less relevant ones.
§ Similarly, the number of references in Line 127 can be limited. Please keep the most relevant and recent ones. The same goes for Lines 322, 398, etc.
§ Most figures (specifically Fig.9) are too complex to understand. Can’t these figures be simplified?
§ The authors mentioned the side effects of intravenous sodium thiosulfate. Would the other routes of administration minimize the side effects?
§ Authors have described the antioxidant, anti-inflammatory, and chelation properties of Sodium thiosulfate in general rather than exploring the deep underlying mechanisms. More associated molecular pathways can be elaborated.
Comments on the Quality of English Language
Minor editing is required to make it publishable.
Author Response
Response to reviewer number 2 round (1).
First authors would like to thank reviewer number 1 for the precious time, effort and knowledge to review this manuscript. Authors responses will appear in blue color in this response and in the resubmitted manuscript.
Comments and Suggestions for Authors
- What could be the possible formulation approaches to enhance the BBB permeability of sodium thiosulfate? Please include them. Please see reviewer number 1 recommendations with delivering STS to the brain, such as nanotechnology-based approaches. These novel approaches could enhance efficacy. Please see lines (32-33) as follows: “Novel strategies such as the future use of nano-technology may be helpful in allowing an increased entry of STS into the brain.”
Also see lines (714-718) as follows: “intratracheal lipopolysaccharide [88]….. To date STS has been administered by intravenous infusions in human individuals and both orally, intraperitoneal and intravenous to preclinical models. In the future, other modes of administration such as the expanding use of nanotechnology or intrathecal administration are potential novel modes of administration in addition to intravenous therapy.”
- Why are there too many keywords? The number can be limited by omitting the less relevant ones. Authors have reduced the keywords from 10 to 7 as in lines 33 and 34
- Similarly, the number of references in Line 127 can be limited. Please keep the most relevant and recent ones. The same goes for Lines 322, 398, etc. Authors agree that the number of references are excessive in each of these mentioned lines; however, we feel that these excessive references are important for the readers to have a better overall understanding of these mechanisms and background for this type of innovative concept regarding the use of STS and wish to retain these references if at all possible.
- Most figures (specifically Fig.9) are too complex to understand. Can’t these figures be simplified? Authors have rebuilt figure 9 and figure 10 in order to provide more simplicity; however, these are complex mechanisms that we are dealing with and figures will need to remain somewhat complex to allow the reader to better understand these complex mechanisms.
- The authors mentioned the side effects of intravenous sodium thiosulfate. Would the other routes of administration minimize the side effects? Currently this question is not answered in the literature that authors have examined. Please note that authors have suggested the use of nanoparticles in the abstract as per reviewer number 1. It is hoped that manuscripts such as this one might increase some interest in exploring these possibilities with greater depth and understanding especially in human studies as well as preclinical models.
- Authors have described the antioxidant, anti-inflammatory, and chelation properties of Sodium thiosulfate in general rather than exploring the deep underlying mechanisms. More associated molecular pathways can be elaborated. Authors agree that each of these mechanisms could be discussed in greater depth and detail; however, the entire Section 2. (2. Antioxidant Effects of STS) was devoted to anti-oxidant effects of STS with a word count of 861;
Sect 3. (3. Anti-Inflammatory Effects of STS with a word count of 974; and
Section 4. (4. Chelation Effects of STS) and in regards to Excess Calcium 4.1, Iron 4.2, and Copper 4.3 with a total word count of 1620.
Therefore, authors feel that they have discussed each of these mechanism with an appropriate amount of depth for the overall submitted manuscript.
To reviewer number 2: please also see the responses to reviewer number 1.
Sincerely with gratitude,
Melvin R Hayden
Submitting author.
